# Molecular oxygen enhances $H_2O_2$ utilization for the photocatalytic conversion of methane to liquid-phase oxygenates

Xiao Sun[1,6], Xuanye Chen[1,6], Cong Fu[1], Qingbo Yu[2], Xu-Sheng Zheng[3], Fei Fang[1], Yuanxu Liu[4], Junfa Zhu [3], Wenhua Zhang [1] & Weixin Huang [1,5] ✉

$H_2O_2$ is widely used as an oxidant for photocatalytic methane conversion to value-added chemicals over oxide-based photocatalysts under mild conditions, but suffers from low utilization efficiencies. Herein, we report that $O_2$ is an efficient molecular additive to enhance the utilization efficiency of $H_2O_2$ by suppressing $H_2O_2$ adsorption on oxides and consequent photogenerated holes-mediated $H_2O_2$ dissociation into $O_2$. In photocatalytic methane conversion over an anatase $TiO_2$ nanocrystals predominantly enclosed by the {001} facets (denoted as $TiO_2${001})-$C_3N_4$ composite photocatalyst at room temperature and ambient pressure, $O_2$ additive significantly enhances the utilization efficiency of $H_2O_2$ up to 93.3%, giving formic acid and liquid-phase oxygenates selectivities respectively of 69.8% and 97% and a formic acid yield of 486 $\mu mol_{HCOOH} \cdot g_{catalyst}^{-1} \cdot h^{-1}$. Efficient charge separation within $TiO_2${001}-$C_3N_4$ heterojunctions, photogenerated holes-mediated activation of $CH_4$ into $\cdot CH_3$ radicals on $TiO_2${001} and photogenerated electrons-mediated activation of $H_2O_2$ into $\cdot OOH$ radicals on $C_3N_4$, and preferential dissociative adsorption of methanol on $TiO_2${001} are responsible for the active and selective photocatalytic conversion of methane to formic acid over $TiO_2${001}-$C_3N_4$ composite photocatalyst.

Methane has been considered as an abundant and promising feedstock for future energy and chemical productions, especially after discovery of large reserves of shale gas and methane hydrate[1,2]. Direct conversion of methane to value-added chemicals has been attracting great interest, however, due to a stable C−H bond, a small polarizability, a high ionization potential and a low electron affinity stability of methane, it remains as a long-standing challenge[3–5]. Harsh reaction conditions, such as high temperatures[6–8] and/or high pressures[9–15], are required for traditional heterogeneous thermocatalytic selective conversion of methane. Recently, photocatalysis has been explored for

selectively converting methane mainly to valuable liquid oxygenates at room temperature and ambient pressure[16–24].

$H_2O_2$ is widely used as an oxidant for photocatalytic selective conversion of methane over oxide-based photocatalysts. Photocatalytic activation of $H_2O_2$ by photo-generated electrons into $\cdot OH$ radicals (0.06 eV vs *RHE*)[25] or $\cdot OOH$ radicals (−0.38 eV vs *RHE*)[26], depending on the conduction band edges of semiconductor photocatalysts, is generally considered as the key step. However, photocatalytic activation of $H_2O_2$ by photo-generated holes into $O_2$ usually occurs facilely[25], which strongly competes and decreases the utilization

[1]Hefei National Research Center for Physical Sciences at the Microscale, iChEM, Key Laboratory of Surface and Interface Chemistry and Energy Catalysis of Anhui Higher Education Institutes, School of Chemistry and Materials Science, University of Science and Technology of China, 230026 Hefei, China. [2]Department of Materials Science and Engineering, Anhui University of Science and Technology, 232001 Huainan, China. [3]National Synchrotron Radiation Laboratory, University of Science and Technology of China, Hefei 230029 Anhui, China. [4]School of Pharmacy, Anhui University of Chinese Medicine, Anhui Academy of Chinese Medicine, Hefei 230012 Anhui, China. [5]Dalian National Laboratory for Clean Energy, Chinese Academy of Sciences, 116023 Dalian, China. [6]These authors contributed equally: Xiao Sun, Xuanye Chen. ✉e-mail: huangwx@ustc.edu.cn

efficiency of $H_2O_2$ for the methane conversion, defined as the ratio of the $H_2O_2$ amount consumed for methane conversion against the total consumed $H_2O_2$ amount. So far, the highest utilization efficiency of $H_2O_2$, in the means of ·OH radicals, was reported as 72.3% in photocatalytic $CH_4$ conversion over a Fenton-type Fe-based catalyst[21]. Adsorption of $H_2O_2$ molecules on photocatalyst surfaces is a prerequisite for occurrences of photocatalytic reactions. Here, we show $O_2$ additive as a general strategy to enhance utilization efficiencies of $H_2O_2$ for the photocatalytic $CH_4$ conversion over oxide-based photocatalysts up to 93.3% by suppressing the $H_2O_2$ adsorption on photocatalyst surfaces and the consequent side reaction of photocatalytic $H_2O_2$ dissociation into $O_2$.

## Results

### Synthesis and structural characterizations

Anatase $TiO_2$ nanocrystals (NCs) predominantly enclosed by the {001} facets (denoted as $TiO_2${001}), the {100} facets (denoted as $TiO_2${100}) and the {101} facets (denoted as $TiO_2${101}) were prepared following well-established recipes[27]. XRD patterns, TEM and HRETM images of as-synthesized various $TiO_2$ NCs (Fig. 1a, Supplementary Fig. 1) agree with those reported previously[27]. $TiO_2$ NCs-$C_3N_4$ composites were prepared by calcination of mixture of calculated amounts of dicyandiamide ($C_2H_4N_4$) and $TiO_2$ NCs in Ar at 550 °C and denoted as $TiO_2$ NCs-$C_3N_4$-x, in which x was the actual $TiO_2$:$C_3N_4$ mole ratio acquired by TGA analysis (Supplementary Fig. 2 and Table 1). TEM, HRTEM and element mapping images (Fig. 1b–d, Supplementary Fig. 3a–c) show that various $TiO_2$ NCs preserve their original morphologies and form smooth anatase $TiO_2$-g-$C_3N_4$ interfaces. We failed to observe clear lattice fringes of g-$C_3N_4$ in the HRTEM images (Supplementary Fig. 3d)

likely due to the strong damage effect of high-energy electron beam on the structure of g-$C_3N_4$, but its presence in the $TiO_2$ NCs-$C_3N_4$ composites is identified by XRD patterns (Supplementary Fig. 3e) and XPS spectra (Supplementary Fig. 3f).

### Photocatalytic performance

$H_2O_2$ barely decomposes at 300 K over various oxides (P25, ZnO, $Fe_2O_3$, $WO_3$, CuO and $V_2O_5$) without Xe light illumination. Under Xe light illumination, $H_2O_2$ decomposition predominantly to $O_2$ occurs slightly in an Ar atmosphere without the presence of oxides but substantially with the presence of oxides (Supplementary Table 2), demonstrating facile occurrence of photogenerated holes-mediated $H_2O_2$ decomposition to $O_2$. Photocatalytic $H_2O_2$ decomposition over various $TiO_2$ NCs was observed dependent on the surface structure. $TiO_2${001} NCs exhibit the lowest photocatalytic activity and $O_2$ selectivity while $TiO_2${101} NCs exhibit the highest (Supplementary Table 3). $C_3N_4$ is poor in photocatalytic $H_2O_2$ decomposition, and comparing corresponding $TiO_2$ NCs, $TiO_2$ NCs-$C_3N_4$ composites exhibit much decreased photocatalytic activity and $O_2$ selectivity (Supplementary Table 3). Interestingly, we found that photocatalytic $H_2O_2$ decomposition over oxides gets greatly suppressed in an $O_2$/Ar atmosphere, together with slight decrease of $O_2$ selectivity; moreover, such an $O_2$ suppress effect varies with the structures of $TiO_2$ NCs and $TiO_2$ NCs-$C_3N_4$ composites (Supplementary Tables 2 and 3). As shown in Fig. 1e, the $H_2O_2$ decomposition percentage/$H_2O_2$ decomposition rate/$O_2$ selectivity are 31.2%/610.9 μmol h$^{-1}$/93.0% over $TiO_2${001} NCs in the Ar atmosphere and decrease to 15.4%/301.5 μmol h$^{-1}$/91.8% in the 10% $O_2$/Ar atmosphere, while they are 20.4%/399.4 μmol h$^{-1}$/89.0% over $TiO_2${001}-$C_3N_4$-0.1 in the Ar

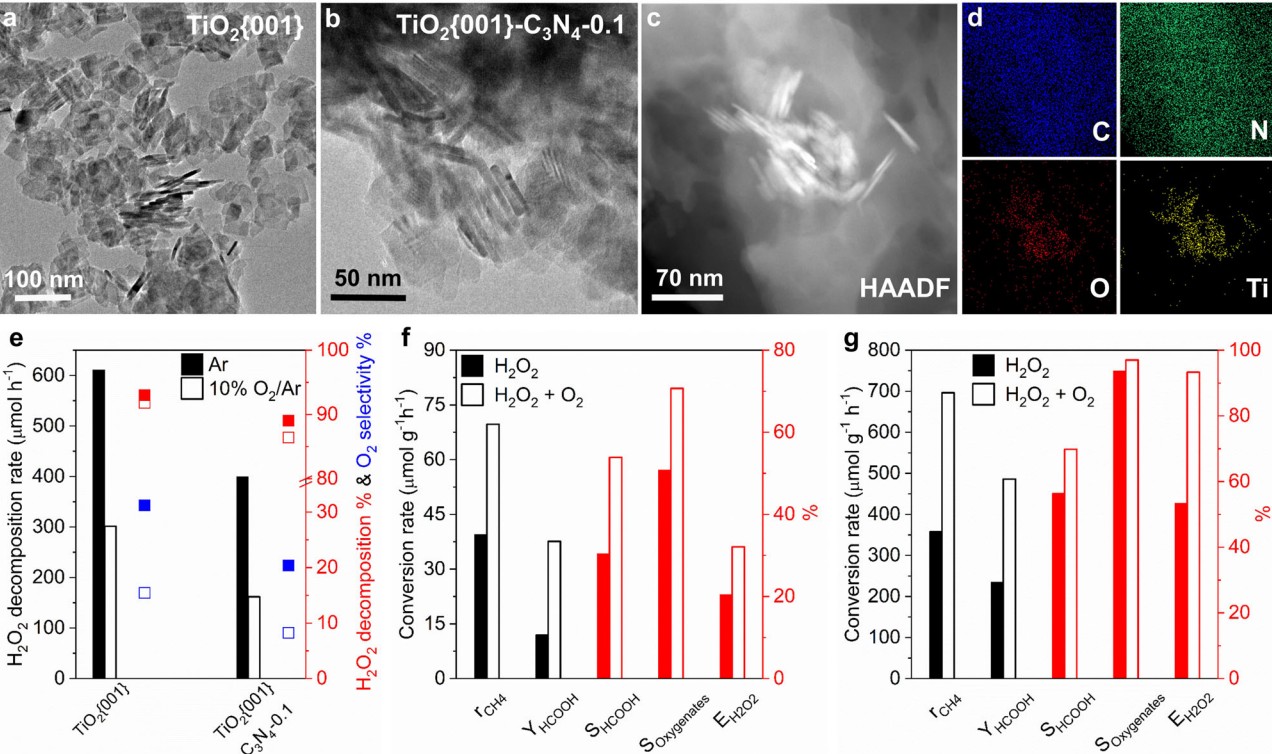

**Fig. 1 | Photocatalysts and photocatalytic performance. a** TEM image of $TiO_2${001}. **b** TEM, (**c**) HAADF and (**d**) element mapping images of $TiO_2${001}-$C_3N_4$-0.1. **e** $H_2O_2$ decomposition rate, $H_2O_2$ decomposition and $O_2$ selectivity of photocatalytic $H_2O_2$ decomposition over $TiO_2${001} and $TiO_2${001}-$C_3N_4$-0.1 under the reaction condition of 165 μL $H_2O_2$ + 20 mL $H_2O$ in Ar or 10%$O_2$/Ar. $CH_4$ conversion rate ($r_{CH4}$), yield($Y_{HCOOH}$) and selectivity ($S_{HCOOH}$) of formic acid, selectivity of oxygenates ($S_{Oxygenates}$), and $H_2O_2$ utilization efficiency ($E_{H2O2}$) of photocatalytic

$CH_4$ conversion over (**f**) 20 mg $TiO_2${001} under the reaction condition of 8% $CH_4$ + 92%Ar + 110 μL $H_2O_2$ + 20 mL $H_2O$ or 8%$CH_4$ + 1.6%$O_2$ + 90.4%Ar + 110 μL $H_2O_2$ + 20 mL $H_2O$ for 5 h and over (**g**) 20 mg $TiO_2${001}-$C_3N_4$-0.1 under the reaction condition of 8%$CH_4$ + 92%Ar + 165 μL $H_2O_2$ + 20 mL $H_2O$ or 8%$CH_4$ + 4% $O_2$ + 88%Ar + 165 μL $H_2O_2$ + 20 mL $H_2O$ for 8 h at 298 K. Source data are provided as a Source Data file.

atmosphere and decrease to 8.26%/161.7 µmol·h$^{-1}$/86.4% in the 10% $O_2$/Ar atmosphere.

The suppress effect of $O_2$ on photocatalytic $H_2O_2$ decomposition into $O_2$ was observed to generally enhance not only $H_2O_2$ utilization efficiency but also $H_2O_2$ conversion, and consequently $CH_4$ conversion in aqueous-phase photocatalytic conversion of methane with $H_2O_2$ using oxide photocatalysts due to the reaction coupling between photocatalytic $H_2O_2$ and $CH_4$ reactions (Supplementary Table 4). Under the studied condition, the $H_2O_2$ utilization efficiency and $CH_4$ conversion with an $O_2$ addition are 1.30–1.78 and 1.4–2.0 times of those without $O_2$ addition, respectively. We then optimized the $O_2$ enhancement effect and photocatalytic performance over $TiO_2$ NCs and $TiO_2$ NCs-$C_3N_4$ composites (Supplementary Tables 5–10), both of which were observed to vary with structures of $TiO_2$ NCs. $TiO_2${001} NCs are more photocatalytic active than $TiO_2${100} and $TiO_2${101} NCs, and the produced liquid-phase oxygenates are $CH_3OH$ and $HCOOH$ over $TiO_2${001} NCs and $CH_3OH$ over $TiO_2${100} and $TiO_2${101} NCs. Over $TiO_2${001} NCs (Fig. 1f), the $O_2$ addition increases the methane conversion rate from 39.5 to 69.7 µmol·$g_{catalyst}^{-1}$·h$^{-1}$, the selectivity of liquid-phase oxygenates and $HCOOH$ respectively from 50.8% to 70.7% and from 30.4% to 53.9%, the $HCOOH$ yield from 12.0 to 37.6 µmol·$g_{catalyst}^{-1}$ h$^{-1}$, and the $H_2O_2$ utilization efficiency from 21.4% to 32.1%. $TiO_2$ NCs-$C_3N_4$–0.1 composites exhibit much better photocatalytic performance and more significant $O_2$ promotion effect than corresponding $TiO_2$ NCs. The produced liquid-phase oxygenates are $CH_3OH$ and $HCOOH$ over $TiO_2${001}-$C_3N_4$–0.1, $CH_3OH$ and $CH_3OOH$ over $TiO_2${100}-$C_3N_4$–0.1, and $CH_3OOH$ over $TiO_2${101}-$C_3N_4$–0.1. Over $TiO_2${001}-$C_3N_4$–0.1 (Fig. 1g), the $O_2$ addition increases the methane conversion rate from 358.5 to 696.3 µmol $g_{catalyst}^{-1}$ h$^{-1}$, the selectivity of liquid-phase oxygenates and $HCOOH$ respectively from 93.7% to 97.0% and from 56.4% to 69.8%, the $HCOOH$ yield from 202.2 to 486 µmol $g_{catalyst}^{-1}$ h$^{-1}$, and the $H_2O_2$ utilization efficiency from 53.4% to 93.3%.

The above results demonstrate an interesting photocatalytic system for efficiently converting $CH_4$ to liquid-phase oxygenates in the presence of $H_2O_2$ and $O_2$ at room temperature and ambient pressure over oxide-based photocatalysts, which presents high $H_2O_2$ utilization efficiencies due to the suppress effect of $O_2$ on photocatalytic $H_2O_2$ decomposition into $O_2$. The best photocatalyst, $TiO_2${001}-$C_3N_4$–0.1, exhibits an unprecedented $H_2O_2$ utilization efficiency of 93.3%, leading to a liquid-phase oxygenates selectivity of 97% and formic selectivity and yield respectively of 69.8% and 486 µmol$_{HCOOH}$·$g_{catalyst}^{-1}$ h$^{-1}$. Its apparent quantum efficiency at 365 nm was measured to be 0.48%.

$TiO_2${001}-$C_3N_4$–0.1 is stable and its performance maintains well within six cycles of photocatalytic activity evaluations (Supplementary Fig. 4). Routine structural characterization results (Supplementary Fig. 5), including XPS, VB-XPS, UV-Vis spectra and photocurrent measurements, show few difference between the as-synthesized and used $TiO_2${001}-$C_3N_4$–0.1 catalysts.

## Reaction mechanism

The carbon balance was calculated above 96.7% for all studied photocatalytic reactions. Blank photocatalytic experiment of photocatalytic reaction in the presence of $TiO_2${001}-$C_3N_4$–0.1 but absence of $CH_4$ in the reactant did not produce detectable C-contained products; meanwhile, using $^{13}CH_4$, all C-contained products only contained $^{13}C$ (Supplementary Fig. 6). Thus, all C-contained products exclusively form from $CH_4$. Initial evolutions of reaction products as a function of reaction time were examined over $TiO_2${001}-$C_3N_4$–0.1 (Supplementary Table 11). At a reaction time of 10 min, $CH_3OOH$, $CH_3OH$ and $HCHO$ were detected, and $CH_3OOH$ was the major product. The $CH_3OOH$, $CH_3OH$ and $HCHO$ productions increased at a reaction time of 30 min, meanwhile, $HCOOH$ and $CH_3CH_2OH$ appeared. As a reaction time of 1 h, the $CH_3OOH$ production decreased and $HCHO$ was not detected, whereas the $CH_3OH$ and $HCOOH$ productions increased greatly and the $CH_3CH_2OH$ production increased slightly, meanwhile,

$CH_3COOH$ emerged. These observations suggest $CH_3OOH$ as the primary product and $CH_3OH$, $HCHO$, $HCOOH$, $CH_3CH_2OH$ and $CH_3COOH$ as the secondary products that are produced sequentially. Moreover, the reaction rate of $HCHO$ seems to be faster than the formation rate.

$^{18}O_2$ and $H_2^{18}O$ were used to trace origins of oxygen atoms in the liquid-phase oxygenate products. $^{18}O_2$ were observed to exert similar enhancement effects on the $H_2O_2$ utilization efficiency to $^{16}O_2$ and to slightly affect the product selectivity (Supplementary Table 12). It is noteworthy that $CH_3OOH$ decomposes completely into $CH_3OH$ during the mass spectroscopy analysis (13). Over $TiO_2${001} NCs (Supplementary Figs. 7, 8), no $^{18}O$-labelled product was detected when $H_2^{18}O$ was used, while $CH_3^{18}OH$ and $HC^{18}O^{16}OH$ were detected with $CH_3^{18}OH$/$CH_3^{16}OH$ and $HC^{18}O^{16}OH$/$HC^{16}O^{16}OH$ ratios respectively of around 0.12 and 0.11 when $^{18}O_2$ was used. Over $TiO_2${001}-$C_3N_4$–0.1 (Fig. 2a–d and Supplementary Fig. 9), only $CH_3C^{18}O^{16}OH$ for $CH_3COOH$ and no other $^{18}O$-labelled oxygenate were detected when $H_2^{18}O$ was used, while $CH_3^{18}OH$, $HC^{18}O^{16}OH$ and $CH_3CH_2^{18}OH$ were detected with $CH_3^{18}OH$/$CH_3^{16}OH$, $HC^{18}O^{16}OH$/$HC^{16}O^{16}OH$ and $CH_3CH_2^{18}OH$/$CH_3CH_2^{16}OH$ ratios respectively of around 0.14, 0.13 and 0.25 when $^{18}O_2$ was used, and $CH_3C^{16}O^{16}OH$ and $CH_3C^{16}O^{18}OH$ were detected for $CH_3COOH$. Therefore, the oxygen atoms in $CH_3OOH$, $CH_3OH$, $HCOOH$ and $CH_3CH_2OH$ are contributed majorly by $H_2O_2$ and minor by $O_2$, but seldom by $H_2O$. Interestingly, $HCOOH$ is formed via $CH_3OH$ oxidation exclusively by $H_2O_2$ whereas $CH_3COOH$ is formed via $CH_3CH_2OH$ oxidation exclusively by $H_2O$, suggesting that they follow different mechanisms. This was further supported by the observations that $HC^{16}O^{16}OH$/$HC^{18}O^{16}OH$ and $CH_3C^{18}O^{16}OH$/$CH_3C^{18}O^{18}OH$ were detected when $^{18}O_2$ and $H_2^{18}O$ were used (Supplementary Fig. 10). Photocatalytic $CH_3CH_2OH$ oxidation with $H_2O$ to $CH_3COOH$ was reported to be mediated by ·OH radicals generated by photogenerated holes-participated activation of $H_2O$, typically occurring in the aqueous solution[28,29]. Thus, photocatalytic reactions to other liquid-phase products occur on the photocatalyst surfaces. Meanwhile, only a tiny amount of $C^{16}O^{18}O$ was detected in the gas phase products while no $C^{18}O$ and $C^{18}O_2$ was detected when $^{18}O_2$ was used for both $TiO_2${001} NCs and $TiO_2${001}-$C_3N_4$–0.1 (Supplementary Fig. 11).

In order to further clarify the role of $O_2$, the $O_2$ concentration in the reactant was increased from 4% (8%$CH_4$ + 4%$O_2$ + 88%Ar + 165 µL $H_2O_2$ + 20 mL $H_2O$) to 12% (8%$CH_4$ + 12%$O_2$ + 80%Ar+165 µL $H_2O_2$ + 20 mL $H_2O$), and the photocatalytic reaction was studied over $TiO_2${001}-$C_3N_4$–0.1 comparatively with $^{16}O_2$ or $^{18}O_2$. Using $^{16}O_2$ or $^{18}O_2$ gave similar $H_2O_2$ utilization efficiencies of around 94% and slightly different $CH_4$ conversion rates and product selectivity (Supplementary Table 13). Using $^{18}O_2$, the $CH_3^{18}OH$/$CH_3^{16}OH$, $HC^{18}O^{16}OH$/$HC^{16}O^{16}OH$ and $CH_3CH_2^{18}OH$/$CH_3CH_2^{16}OH$ ratios in the liquid-phase products were measured respectively as around 0.19, 0.17 and 0.22 (Supplementary Figs. 12–14), similar to the case of the reactant with 4% $O_2$; however, $C^{18}O$ and $C^{18}O_2$ were detected and the fraction of $C^{16}O^{18}O$ in $CO_2$ is much larger than that of $C^{16}O^{16}O$, different from the case of the reactant with 4% $O_2$. Therefore, during photocatalytic aqueous-phase $CH_4$ conversion in the presence of $H_2O_2$ and $O_2$, $CH_4$ preferentially reacts with $H_2O_2$ to produce liquid-phase oxygenates, while $O_2$ acts mainly as a promoter to enhance $H_2O_2$ utilization efficiency and consequently $CH_4$ conversion, and minorly as a reactant.

Using 5, 5-dimethyl-1-pyrroline N-oxide (DMPO) as the radical trapping agent, in situ EPR was used to probe radicals generated by photo-induced activation of various reactants. As shown in Fig. 2e and Supplementary Fig. 15, under UV light illumination, $H_2O$ is activated to ·OH radicals[30] by photogenerated holes ($h^+$) over various $TiO_2$ NCs and $TiO_2$ NCs-$C_3N_4$–0.1 composites, which barely changes upon the addition of $O_2$. ·$O_2^-$ radicals formed by $O_2$ activation with photogenerated electrons ($e^-$) can not be observed in ESR spectra due to the instability in the aqueous solution, but their formation is evidenced by in situ ESR spectra in the methanol solution[31] (Supplementary Fig. 16). Over $TiO_2$ NCs, the ·OH radical signal grows slightly upon the addition of $H_2O_2$

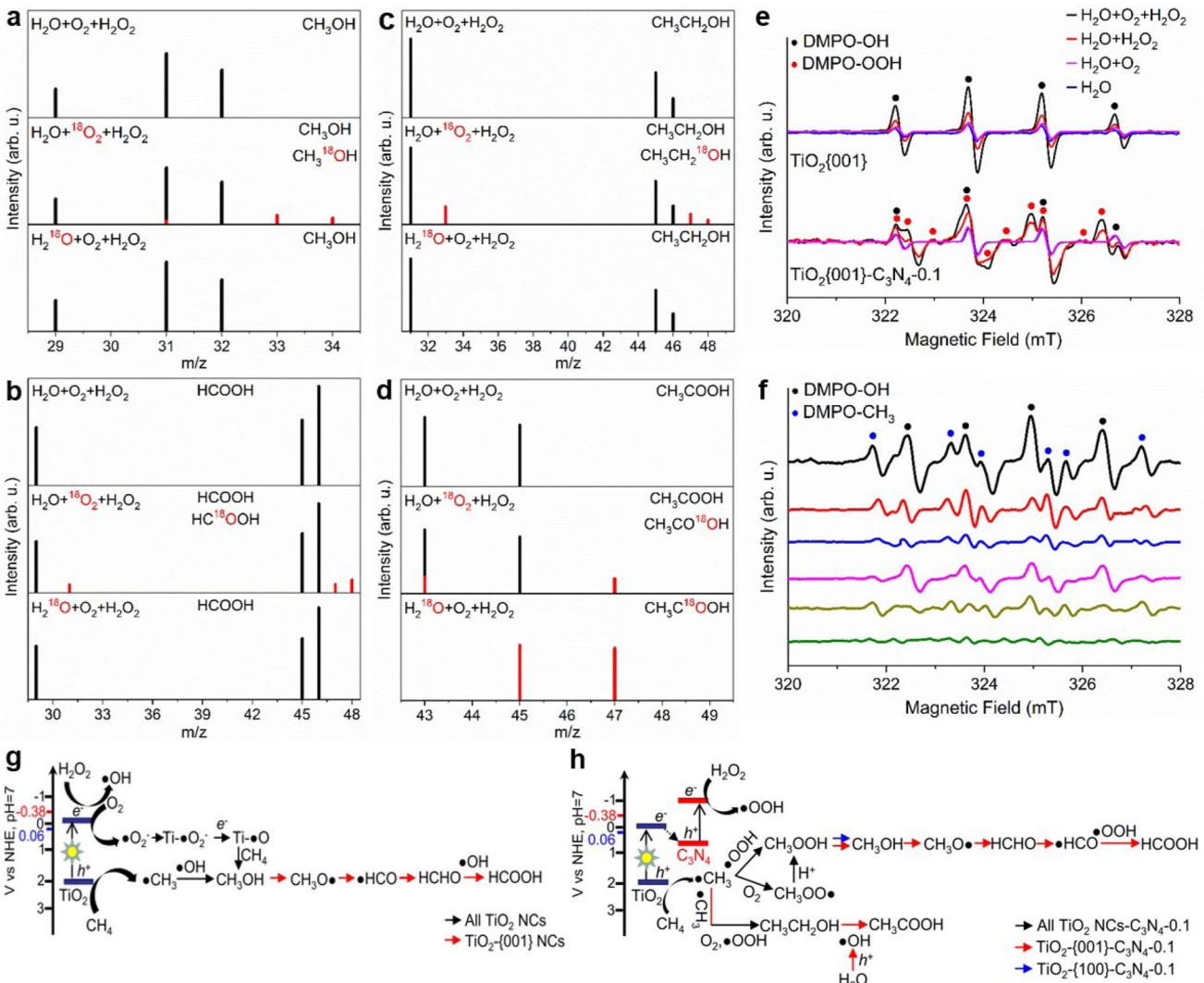

**Fig. 2 | Reaction mechanism.** Mass spectra of (**a**) methanol, (**b**) formic acid, (**c**) ethanol and (**d**) acetic acid formed during photocatalytic $CH_4$ conversion over $TiO_2\{001\}$-$C_3N_4$−0.1 under the reaction condition of 8%$CH_4$ + 4%$O_2$ + 88%Ar + 165 µL $H_2O_2$ + 20 mL $H_2O$, 8%$CH_4$ + 4%$^{18}O_2$ + 88%Ar + 165 µL $H_2O_2$ + 20 mL $H_2O$, or 8% $CH_4$ + 4%$O_2$ + 88%Ar + 165 µL $H_2O_2$ + 20 mL $H_2^{18}O$. Photocatalyst amount: 20 mg; reaction temperature: 25 °C; reaction time: 8 h. **e** In situ ESR spectra of $H_2O$, $H_2O$ + $O_2$, $H_2O$ + $H_2O_2$ and $H_2O$ + $O_2$ + $H_2O_2$ solutions under UV light illumination at 298 K in the presence of DMPO over $TiO_2\{001\}$ NCs and $TiO_2\{001\}$-$C_3N_4$−0.1. **f** In situ

ESR spectra of $CH_4$ + $H_2O$ mixture under UV light illumination at 298 K in the presence of DMPO over $TiO_2\{101\}$ (olive), $TiO_2\{100\}$ (dark yellow) and $TiO_2\{001\}$ (magenta) NCs, $TiO_2\{101\}$-$C_3N_4$−0.1 (blue), $TiO_2\{100\}$-$C_3N_4$−0.1 (red) and $TiO_2\{001\}$-$C_3N_4$−0.1 (black) composites. Schematic diagrams of proposed dominant photocatalytic aqueous-phase $CH_4$ reaction paths to liquid-phase oxygenates in the presence of $H_2O_2$ and $O_2$ over (**g**) $TiO_2$ NCs and (**h**) $TiO_2$ NCs-$C_3N_4$. The 0.06 eV and −0.38 eV refer to the redox potential of $H_2O_2$ activated to ·OH radicals and ·OOH radicals at pH = 7. Source data are provided as a Source Data file.

and greatly upon the co-addition of $H_2O_2$ and $O_2$. Over $TiO_2$ NCs-$C_3N_4$−0.1 composites, the ·OH radical signal does not vary upon the addition of $H_2O_2$, while the ·OOH radical signal[25,26] appears, and its intensity increases greatly upon the co-addition of $H_2O_2$ and $O_2$. Thus, under UV light illumination, in addition to the $h^+$-mediated decomposition into $O_2$, $H_2O_2$ undergoes the $e^-$-mediated activation into ·OH radicals over $TiO_2$ NCs and ·OOH radicals over $TiO_2$ NCs-$C_3N_4$−0.1 composites. The $e^-$-mediated formation of dominant ·OOH radicals but few ·OH radicals over $TiO_2$ NCs-$C_3N_4$−0.1 composites indicates that $e^-$ for $H_2O_2$ activation is located on the conduction band mainly of $C_3N_4$ but seldom of $TiO_2$, pointing to efficient interfacial transfer of $e^-$ from the conduction band of $TiO_2$ to the conduction band of $C_3N_4$. When $CH_4$ was introduced to the aqueous solutions containing $TiO_2$ NCs or $TiO_2$ NCs-$C_3N_4$−0.1 composites under UV light illumination (Fig. 2f), ·$CH_3$ radicals[22,26], in addition to ·OH radicals, were detected. They greatly grew when isopropanol was added to quench ·OH radicals (Supplementary Fig. 17), but could not be detected in the presence of

$H_2O_2$ and $O_2$ when $h^+$ was quenched using methanol (Supplementary Fig. 18). Thus, photocatalytic $CH_4$ activation to ·$CH_3$ radicals is mainly mediated by $h^+$, instead of by·OOH, ·OH and ·$O_2^-$ radicals.

Based on the above isotope-labelled and ESR results, photocatalytic $CH_4$ conversion with $H_2O_2$ is initiated by the reaction of $h^+$-generated ·$CH_3$ with $e^-$-generated ·OH to $CH_3OH$ over $TiO_2$ NCs (Fig. 2g) and with $e^-$-generated ·OOH to $CH_3OOH$ over $TiO_2$ NCs-$C_3N_4$−0.1 composites (Fig. 2h). The addition of $O_2$ opens up minor reaction pathways, including the reaction of $h^+$-generated ·$CH_3$ with $O_2$ to $CH_3OO$· radicals that facilely transform to $CH_3OOH$[22,32] and the reaction of $CH_4$ with Ti−O· formed by $e^-$-mediated ·$O_2^-$ reactions on $TiO_2$ surfaces directly to $CH_3OH$[33]. We consider that the ·$CH_3$ + $O_2$ reaction occurs mainly for $TiO_2$ NCs-$C_3N_4$−0.1 composites due to the lack of enough $e^-$ on the $TiO_2$ components while the $CH_4$ + Ti−O· reaction occurs mainly for $TiO_2$ NCs due to the absence of $CH_3OOH$ in the products. Moreover, the addition of $O_2$ greatly enhances the intensities of ·OOH radicals over $TiO_2$ NCs-$C_3N_4$−0.1 composites and

·OH radicals over $TiO_2$ NCs formed by the $e^-$-mediated $H_2O_2$ activation, and consequently the photocatalytic $CH_4$ conversions. Since the presence of $O_2$ efficiently suppresses the $h^+$-mediated $H_2O_2$ decomposition to $O_2$ under UV light illumination, the enhancement effect of $O_2$ on ·OH and ·OOH generations from photocatalytic $H_2O_2$ activation is probably due to $O_2$-suppressed $h^+$-mediated $H_2O_2$ decomposition to $O_2$ rather than $O_2$-promoted $e^-$-mediated $H_2O_2$ decomposition to ·OH and ·OOH radicals. $O_2$ does not compete with $H_2O_2$ for $h^+$ that is localized on the $TiO_2$ surface, thus $O_2$ likely suppresses $H_2O_2$ adsorption on $TiO_2$, instead of reaction of adsorbed $H_2O_2$ with $h^+$, to suppress the $h^+$-mediated $H_2O_2$ decomposition to $O_2$. Both $TiO_2$ NCs and $TiO_2$ NCs-$C_3N_4$−0.1 composites exhibit $TiO_2$ facet-dependent intensities of various radicals. The ·OH radicals are strongest over $TiO_2${001} NCs among all $TiO_2$ NCs and the ·OOH radicals are strongest over $TiO_2${001}-$C_3N_4$−0.1 composite among all $TiO_2$ NCs-$C_3N_4$ composites (Supplementary Fig. 19). The ·$CH_3$ radicals are strongest over $TiO_2${100} NCs among various $TiO_2$ NCs and over $TiO_2${001}-$C_3N_4$−0.1 composite among various $TiO_2$ NCs-$C_3N_4$ composites (Supplementary Fig. 20). Meanwhile, $TiO_2$ NCs-$C_3N_4$ composites exhibit more reactive radicals than corresponding $TiO_2$ NCs. These results are consistent with the results of photocatalytic activity.

The band structures of various $TiO_2$ NCs and $TiO_2$ NCs-$C_3N_4$−0.1 photocatalysts were determined using UV–vis spectra and valence band XPS spectra (Supplementary Fig. 21). $TiO_2$ NCs-$C_3N_4$−0.1 exhibits smaller band gaps than corresponding the $TiO_2$ NCs, suggesting stronger capacity for light absorption and charge generation. The conduction band edges of $TiO_2$ NCs and $TiO_2$ NCs-$C_3N_4$ composites were measured to be −0.14~−0.34 and −0.41~−0.47 vs *RHE*, respectively, consistent with the experimental observations that $H_2O_2$ undergoes the $e^-$-mediated activation into ·OH radicals over $TiO_2$ NCs and ·OOH radicals over $TiO_2$ NCs-$C_3N_4$−0.1 composites (Fig. 2g, h). ESR spectra (Supplementary Fig. 22a) show that $TiO_2$ NCs-$C_3N_4$−0.1 exhibit much lower densities of $F^+$ and $Ti^{3+}$ defects than $TiO_2$ NCs and the defect density follows an order of $TiO_2${101} > $TiO_2${100} > $TiO_2${001} > $TiO_2${101}-$C_3N_4$−0.1 > $TiO_2${100}-$C_3N_4$−0.1 > $TiO_2${001}-$C_3N_4$−0.1. Accordingly, PL spectra (Supplementary Fig. 22b) show that the PL peak arising from the recombination of photoexcited electrons and holes displays an intensity order of $TiO_2${101} > $TiO_2${100} > $TiO_2${001} > $TiO_2${101}-$C_3N_4$−0.1 > $TiO_2${100}-$C_3N_4$−0.1 > $TiO_2${001}-$C_3N_4$−0.1. EIS spectra of various $TiO_2$ NCs and $TiO_2$ NCs-$C_3N_4$−0.1 photocatalysts were also measured, in which a smaller radius represents a low charge transfer resistance. All photocatalysts exhibit semicircle EIS spectra (Supplementary Fig. 22c), and the semicircle radius and consequently the charge transfer resistance follow an order of $TiO_2${101} > $TiO_2${100} > $TiO_2${001} > $TiO_2${101}-$C_3N_4$−0.1 > $TiO_2${100}-$C_3N_4$−0.1 > $TiO_2${001}-$C_3N_4$−0.1. ESR, PL and EIS are all bulk-sensitive characterization techniques, and their characterization results show that $TiO_2$ NCs-$C_3N_4$−0.1 exhibit higher charge separation and transfer efficiencies than corresponding $TiO_2$ NCs and that $TiO_2${001} is the best of various $TiO_2$ NCs while $TiO_2${001}-$C_3N_4$−0.1 is the best of $TiO_2$ NCs-$C_3N_4$ composite photocatalysts, consistent with the photocatalytic activity results.

NEXAFS acquired in a mode of total electron yield is a surface sensitive technique to probe the density of states of the orbitals involved in the electron transitions. UV light illumination excites electrons from the valence band to the conduction band, which consequently changes the density of states of the involved orbitals. We thus measured Ti L-edge, O K-edge, N K-edge and C K-edge NEXAFS spectra under dark and UV light illumination conditions of various samples (Fig. 3a–d, Supplementary Figs. 23, 24). The valence and conduction bands of $TiO_2$ consist of the O 2*p* and Ti 3*d* orbitals, respectively, and the Ti L-edge and O K-edge NEXAFS features arise from the Ti 2*p*→3*d* and O 1*s*→2*p* electron transitions, respectively. The valence and conduction bands of $C_3N_4$ consist of the N 2*p* and C 2*p* orbitals, respectively, and the N K-edge and C K-edge NEXAFS features arise from the N 1*s*→2*p* and C 1*s*→2*p* electron transitions, respectively.

$TiO_2$ NCs-$C_3N_4$ composites exhibit enhanced Ti L-edge and O K-edge features than corresponding $TiO_2$ NCs but weakened C K-edge and N K-edge NEXAFS features than $C_3N_4$. This indicates an occurrence of $TiO_2$→$C_3N_4$ electron transfer within $TiO_2$ NCs-$C_3N_4$ composites, which decreases the electron density on $TiO_2$ but increases the electron density of $C_3N_4$. Using the Ti L-edge and C K-edge NEXAFS features as examples (Fig. 3e), $TiO_2${001}-$C_3N_4$ composite exhibits the largest intensity variations of both Ti-L edge and C-K edge absorption features among all $TiO_2$ NCs-$C_3N_4$ composites, demonstrating the most extensive electron transfer from $TiO_2${001} NCs to $C_3N_4$.

UV light illumination excites electrons from the valence bands of $TiO_2$ or $C_3N_4$ to the conduction bands, and consequently results in weakened Ti L-edge NEXAFS features of $TiO_2$ NCs, enhanced O K-edge NEXAFS features of $TiO_2$ NCs and $TiO_2$ NCs-$C_3N_4$−0.1 composites, and weakened C K-edge and enhanced N K-edge NEXAFS features of $C_3N_4$ and $TiO_2$ NCs-$C_3N_4$−0.1 composites. But $TiO_2$ NCs-$C_3N_4$−0.1 composites exhibit stronger Ti L-edge NEXAFS features under UV light illumination than under dark condition. This supports the formations of Z-scheme $TiO_2$-$C_3N_4$ heterojunctions within $TiO_2$ NCs-$C_3N_4$−0.1 composites[34], in which the photogenerated electrons on the conduction band of $TiO_2$ (Ti 3*d* orbital) efficiently transfer to the valence band of $C_3N_4$ (N 2*p* orbital) and recombine with photogenerated holes therein (Fig. 2h). Moreover, the total transferred electrons from the conduction band of $TiO_2$ are more than the photogenerated electrons, likely due to a large number of photogenerated holes in the valence band of $C_3N_4$, which results in a less-occupied Ti 3*d* orbital and consequently a stronger Ti L-edge NEXAFS features of $TiO_2$-$C_3N_4$−0.1 composites under UV light illumination than under dark condition. Figure 3f presents the ratios ($I_{UV}/I_{dark}$) of Ti L-edge, O K-edge, N K-edge and C K-edge NEXAFS features of various photocatalysts under UV light illumination against in dark, whose deviations from the unity reflect the photogenerated charges on the photocatalyst surfaces. Much larger concentrations of photogenerated electrons and holes are present on $TiO_2$ surfaces than on $C_3N_4$ surface, suggesting more efficient charge separation and migration to surface within $TiO_2$ NCs. $C_3N_4$ surface exhibits similar concentrations of photogenerated electrons and holes while $TiO_2$ surfaces exhibit larger concentrations of photogenerated holes than of photogenerated electrons. $TiO_2$ NCs-$C_3N_4$−0.1 composite surfaces exhibit slightly smaller concentrations of photogenerated holes than corresponding $TiO_2$ NCs surfaces but larger concentrations of photogenerated charges than $C_3N_4$ surfaces. Thus, the Z-scheme $TiO_2$-$C_3N_4$ heterojunctions within $TiO_2$ NCs-$C_3N_4$−0.1 composites contribute to the charge separation and migration to surface over $C_3N_4$ component more than over $TiO_2$ component. Among various $TiO_2$ NCs or $TiO_2$ NCs-$C_3N_4$−0.1 composites, the photocatalysts consisting $TiO_2${001} NCs exhibit the largest concentrations of photogenerated charges on the surfaces, leading to the largest concentrations of ·OH radicals over $TiO_2${001} NCs, ·OOH and ·$CH_3$ radicals over $TiO_2${001}-$C_3N_4$ composite. However, $TiO_2${100} NCs, instead of $TiO_2${001} NCs, exhibit the largest concentration of ·$CH_3$ radicals, which is likely relevant to the adsorption behaviors of $CH_4$ on various photocatalysts. The adsorption heats of $CH_4$ were measured similar for various $TiO_2$ NCs (16.8–17.7 kJ/mol) or $TiO_2$ NCs-$C_3N_4$−0.1 composites (11.1–14.5 kJ/mol) (Supplementary Figs. 25–27), while the adsorption amounts followed orders of $TiO_2${100} > $TiO_2${101} > $TiO_2${001} and of $TiO_2${001}-$C_3N_4$ > $TiO_2${100}-$C_3N_4$ > $TiO_2${101}-$C_3N_4$.

Various $TiO_2$ NCs and $TiO_2$ NCs-$C_3N_4$−0.1 composites show not only $TiO_2$ facet-dependent activity but also $TiO_2$ facet-dependent selectivity in photocatalytic $CH_4$ conversion with $H_2O_2$ or $H_2O_2 + O_2$. The photocatalysts with low photocatalytic activity exhibit low selectivity toward the liquid-phase products because more oxidizing radicals are available to eventually convert the liquid-phase intermediates to $CO_2$. $TiO_2${001} NCs and $TiO_2${001}-$C_3N_4$−0.1 composite exhibit the highest photocatalytic activity and consequently the highest photocatalytic selectivity toward the liquid-phase products among

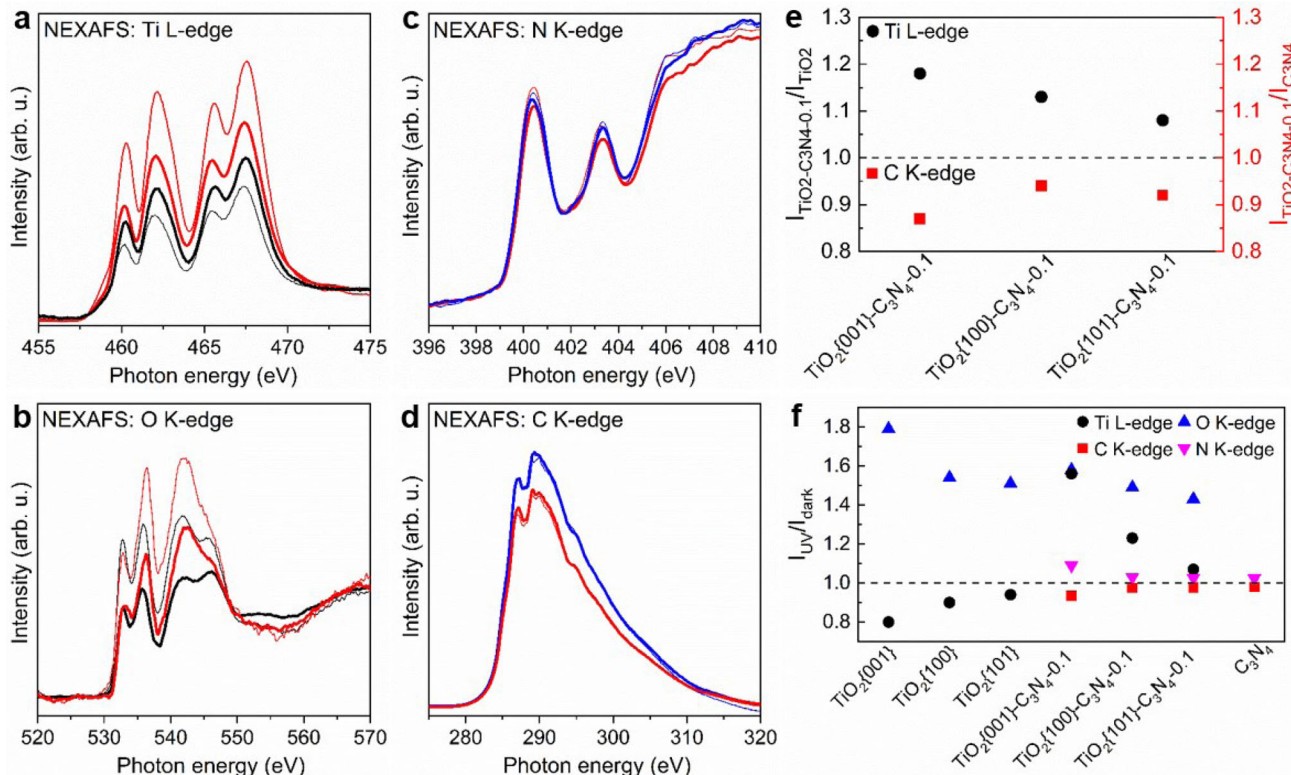

**Fig. 3 | Interfacial charge transfer.** a Ti L-edge, (**b**) O K-edge, (**c**) N K-edge and (**d**) C K-edge NEXAFS spectra of $TiO_2${001} NCs (black line), $TiO_2${001} NCs-$C_3N_4$−0.1 composite (red line) and $C_3N_4$ (blue line) in dark (thick line) and under UV light illumination (thin line). **e** Ti L-edge intensity ratios of $TiO_2$ NCs-$C_3N_4$−0.1 composites against corresponding $TiO_2$ NCs and C K-edge intensity ratios of $TiO_2$ NCs-$C_3N_4$−0.1 composites against $C_3N_4$. **f** Intensity ratios of Ti L-edge, O K-edge, C K-edge and N K-edge features of various photocatalysts under UV light illumination against in dark. Source data are provided as a Source Data file.

various $TiO_2$ NCs and various $TiO_2$ NCs-$C_3N_4$−0.1 composites, respectively. Moreover, distributions of liquid-phase products vary with the $TiO_2$ facets. Especially, HCOOH is the major liquid-phase product for the photocatalysts containing $TiO_2${001} NCs, but is barely observed for the photocatalysts containing $TiO_2${101} or $TiO_2${100} NCs. In situ DRIFTS spectra were used to explore surface reaction mechanisms of photocatalytic $CH_4$ conversion with $H_2O_2$ + $O_2$ over $TiO_2$ NCs-$C_3N_4$−0.1 composites (Fig. 4). The observed vibrational bands (Supplementary Table 14) were assigned based on in situ DRIFTS spectra of $CH_3OH$ and HCOOH adsorption on various $TiO_2$ NCs (Supplementary Fig. 28) and previous reports[35–39]. As the photocatalytic reaction prolongs over $TiO_2${001}-$C_3N_4$−0.1 composite (Fig. 4a), the vibrational features of adsorbed $CH_3$ (1473 $cm^{-1}$), $CH_2$ (1445 $cm^{-1}$), $CH_3OH$ (1019 and 1092 $cm^{-1}$), $CH_3O$ (1042 and 1156 $cm^{-1}$), $CH_2O$ (1712 $cm^{-1}$), HCOO (1526, 1556 and 1564 $cm^{-1}$), HCOOH (1664 $cm^{-1}$) and carbonates (1504 and 1592 $cm^{-1}$) species and gaseous HCOOH (1760 and 1782 $cm^{-1}$) emerge and grow at the expense of gaseous $CH_4$ (1304 $cm^{-1}$). These results directly evidence the occurrences of photocatalytic oxidation of $CH_4$ to $CH_3OH$ via the $CH_3$ intermediate and further to HCOOH via the $CH_3O$, $CH_2O$ and HCO (in the form of $HCOO_{TiO_2}$) intermediates, as schematically shown in Fig. 2g, h. Although the carbonate intermediates were observed, no signals of CO or $CO_2$ appeared, indicating that the carbonate intermediates are very stable on $TiO_2${001}-$C_3N_4$ composite. Comparing $TiO_2${001}-$C_3N_4$−0.1 composite, $TiO_2${100}-$C_3N_4$−0.1 and $TiO_2${101}-$C_3N_4$−0.1 composites exhibit very different in situ DRIFTS spectra (Fig. 4b). The gaseous $CH_4$ consumptions and the $CH_3OH$(a) formation are greatly smaller over $TiO_2${100}-$C_3N_4$−0.1 and $TiO_2${101}-$C_3N_4$−0.1 composites than over $TiO_2${001}-$C_3N_4$−0.1 composite. Meanwhile, only very minor vibrational features of surface intermediates appear whereas obvious vibrational features of gaseous CO (2135 and 2170 $cm^{-1}$) and $CO_2$ (2340 and 2360 $cm^{-1}$) emerge over $TiO_2${100}-$C_3N_4$−0.1 and $TiO_2${101}-$C_3N_4$−0.1 composites, respectively.

These in situ DRIFTS results are consistent with the photocatalytic reaction data that $TiO_2${001}-$C_3N_4$−0.1 composite are much more photocatalytic active and selective toward the liquid-phase products in photocatalytic $CH_4$ conversion with $H_2O_2$ + $O_2$ than $TiO_2${100}-$C_3N_4$−0.1 and $TiO_2${101}-$C_3N_4$−0.1 composites.

## Theoretical calculations
DFT calculations were carried out to understand $O_2$-suppressed photocatalytic $H_2O_2$ decomposition to $O_2$ and facet-dependent photocatalytic selectivity of $CH_4$. Since both photocatalytic $H_2O_2$ decomposition to $O_2$ and photocatalytic $CH_4$ conversion are mediated by photogenerated holes located predominantly on $TiO_2$ NCs, thus we considered $TiO_2$ facets, but not $TiO_2$-$C_3N_4$ interfaces, during the DFT calculations. As reported previously[35,40–44], the anatase $TiO_2$(001) surface exposed on $TiO_2${001} NCs exhibits a typical reconstructed (001)-(1 × 4) surface with fourfold-coordinated Ti cations ($Ti_{4c}$) at the (1 × 4) added row, fivefold-coordinated Ti cations ($Ti_{5c}$) at the (1 × 1) basal surface and twofold-coordinated O anions ($O_{2c}$), the anatase $TiO_2$(100) surface exposed on $TiO_2${100} NCs exhibits a typical reconstructed (1 × 2) surface with the $Ti_{5c}$, $O_{2c}$ and threefold-coordinated O ($O_{3c}$) sites, and the anatase $TiO_2$(101) surface exposed on $TiO_2${101} NCs exhibits a (1 × 1) unreconstructed surface with the $Ti_{5c}$, $O_{2c}$ and $O_{3c}$ sites (Supplementary Fig. 29). The $Ti_{4c}$ sites on $TiO_2$(001) surface show much stronger adsorption ability than the $Ti_{5c}$ sites on $TiO_2$ (001), (100) and (101) surfaces. As shown in Fig. 5a and Supplementary Fig. 30, the adsorption energy of $H_2O_2$ is −1.46, −0.80 and −0.77 eV on $TiO_2$ (001), (100) and (101) surfaces, respectively, and greatly decreases to −0.53, −0.18 and −0.10 eV on $O_2$-covered $TiO_2$ (001), (100) and (101) surfaces, respectively. The adsorption energy of $O_2$ is −0.49, −0.18 and −0.14 eV on $TiO_2$ (001), (100) and (101) surfaces, respectively (Fig. 5b and Supplementary Fig. 31). These DFT calculation results demonstrate that $O_2$ is capable of weakening $H_2O_2$ adsorption on $TiO_2$

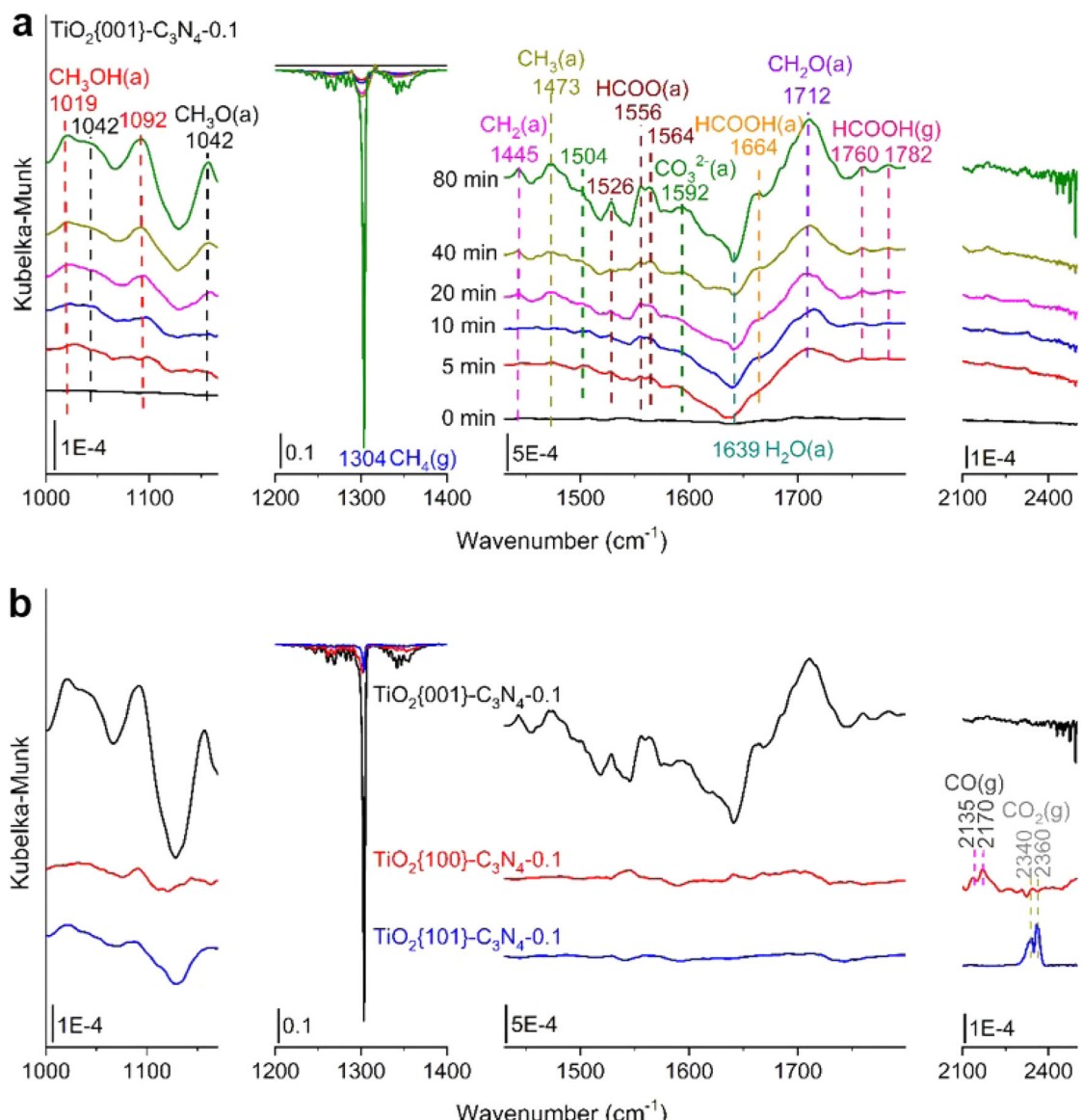

**Fig. 4 | In situ characterization. a** In situ DRIFTS spectra of photocatalytic CH₄ conversion at 298 K under different light irradiation times over TiO₂{001}-C₃N₄−0.1 with DRIFTS spectra prior to UV light illumination as the background spectra. **b** In situ DRIFTS spectra of photocatalytic CH₄ conversion at 298 K under light irradiation for 80 min over TiO₂{001}-C₃N₄−0.1, TiO₂{100}-C₃N₄−0.1 and TiO₂{101}-C₃N₄−0.1 with DRIFTS spectra prior to UV light illumination as the background spectra. Source data are provided as a Source Data file.

to suppress the $h^+$-mediated $H_2O_2$ decomposition to $O_2$. The strongest adsorption of $O_2$ on TiO₂(001) surface exerts the strongest suppress effect on $H_2O_2$ decomposition to $O_2$ on TiO₂{001} NCs. CH₄ adsorption on TiO₂ (001), (100) and (101) surfaces are very weak with an adsorption energy of −0.17, −0.03 and −0.04 eV (Supplementary Fig. 32). Adsorption energy of CH₃OOH on TiO₂ (001), (100) and (101) surfaces is −0.69, −0.22 and −0.04 eV, respectively (Fig. 5c and Supplementary Fig. 33). CH₃OH adsorbs both molecularly and dissociatively with adsorption energy respectively of −0.84 and −1.69 eV on TiO₂ (001) surface, −0.57 and −0.16 eV on TiO₂ (100) surface, −0.49 and −0.65 eV on TiO₂(101) surface (Fig. 5d and Supplementary Fig. 34). The calculated adsorption energies of various liquid-phase products on TiO₂ (001), (100) and (101) surfaces are consistent with the experimentally observed different selectivity toward liquid-phase products in photocatalytic CH₄ conversion over TiO₂ {001}, {100} and {101} NCs, suggesting that desorption of various products from TiO₂ surface play a key role in determining the selectivity. Preferential dissociation of produced CH₃OH on TiO₂{001} NCs and TiO₂{001}-C₃N₄−0.1

composite forms methoxy species which is further photooxidized to HCOOH (Fig. 2g, h), leading to the experimental results that HCOOH is the major liquid-phase product. The very weak adsorption of produced CH₃OOH on TiO₂(101) surface makes it as the sole liquid-phase product over TiO₂{101}-C₃N₄−0.1 composite.

## Discussion
Therefore, $O_2$ is a general and efficient molecular additive to suppress $H_2O_2$ adsorption on oxide photocatalysts and consequently photogenerated holes-mediated $H_2O_2$ decomposition to $O_2$ during photocatalytic reactions. Such a suppress effect, together with efficient charge separation within TiO₂{001}-C₃N₄ heterojunctions, photogenerated holes-mediated activation of CH₄ into ·CH₃ radicals on TiO₂{001} and photogenerated electrons-mediated activation of $H_2O_2$ into ·OOH radicals on C₃N₄, and preferential dissociative adsorption of methanol on TiO₂{001}, leads to an unprecedented high $H_2O_2$ utilization efficiency of 93.3% and highly active and selective to liquid-phase oxygenates with formic acid as the major product during

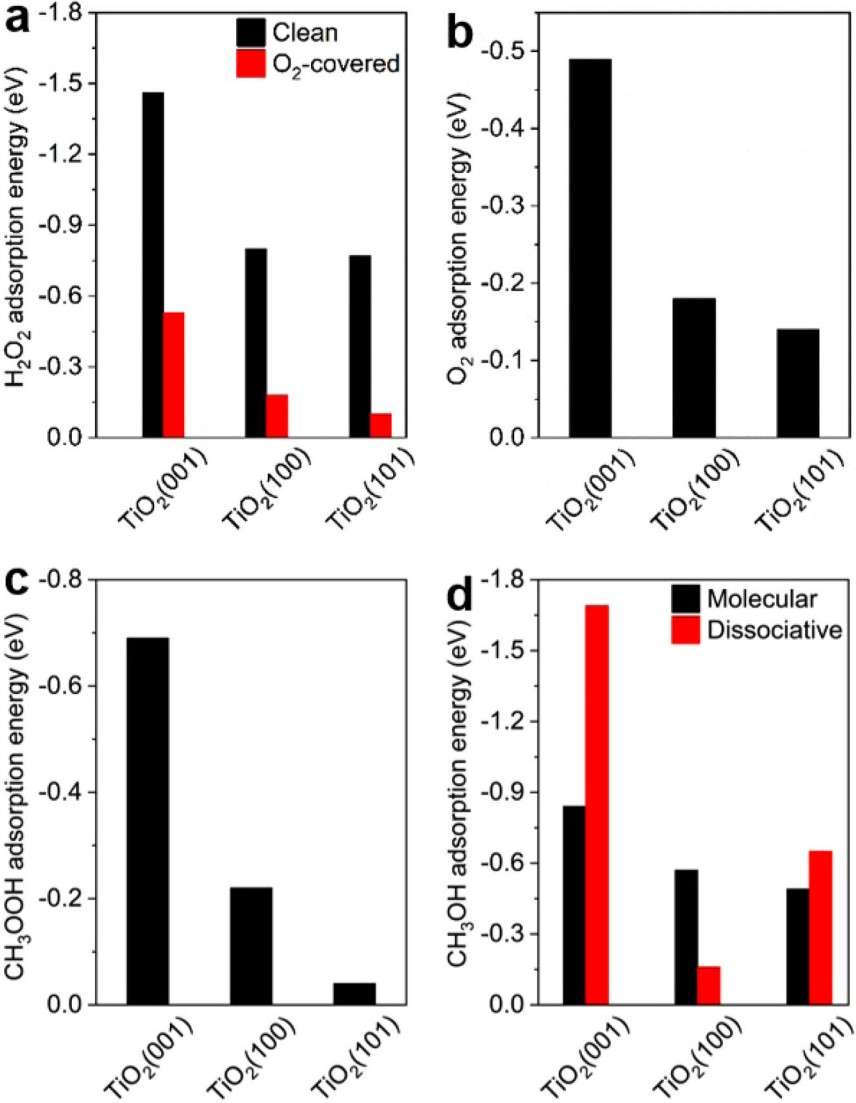

**Fig. 5 | DFT calculations.** Calculated adsorption energies of (**a**) $H_2O_2$ on clean and $O_2$-covered $TiO_2$ (001), (100) and (101) surfaces, (**b**) $O_2$, (**c**) $CH_3COOH$ and (**d**) molecular and dissociative $CH_3OH$ adsorption on $TiO_2$ (001), (100) and (101) surfaces. Source data are provided as a Source Data file.

photocatalytic conversion of methane with $H_2O_2$ and $O_2$. $H_2O_2$ production is known as an environment-unfriendly and economic-costly process[45], therefore, our findings point to co-use of $H_2O_2$ and $O_2$ in photocatalytic oxidation reactions over oxide-based photocatalysts as a promising strategy to achieve high $H_2O_2$ utilization efficiency and excellent photocatalytic performance.

## Methods
### Materials
$H_2O_2$ aqueous solution (20 wt.%), HF aqueous solution (40 wt.%), acetate, acetic acid, methanol, isopropanol, $Ti(OBu)_4$, $K_2TiO(C_2O_4)_2$, P25, ZnO, $Fe_2O_3$, $WO_3$ and $V_2O_5$ were all with the analytical grade and purchased from Sinopharm Chemical Reagent Co. CuO (≥99%), dicyandiamide (≥98%), pentane-2,4-dione (≥98%), 5, 5-dimethyl-1-pyrroline N-oxide (DMPO) (≥97%) and 3-(trimethylsilyl)−1-propane-sulfonic acid sodium salt (DSS) (≥97%) were purchased from Sinopharm Chemical Reagent Co. Reactants of $CH_4$ (8%) + $O_2$(4%) + Ar (88%) and $CH_4$ (8%) + $O_2$ (4%) + $O_2$ (10%) + Ar (78%) were purchased from Nanjing Shang Yuan Industry Factory. $^{13}CH_4$ ($^{13}C$ enrichment > 99%atom), $^{18}O_2$ ($^{18}O$ enrichment ≥ 98%atom) and $H_2^{18}O$ ($^{18}O$ enrichment ≥ 98%atom)

were purchased from Wuhan Newradar Gas Co. All chemicals and gases were used as received.

### Catalyst synthesis
$TiO_2$ NCs predominantly exposing different types of facets were prepared following previous procedures[27].

Synthesis of anatase $TiO_2\{001\}$ NCs: typically, 25 mL $Ti(OBu)_4$ and 3 mL HF aqueous solution (40 wt%) were mixed under stirring at RT (Caution: Hydrofluoric acid (HF) is extremely corrosive and a contact poison, and it should be handled with extreme care! Hydrofluoric acid solution is stored in Teflon containers in use.). The solution was then transferred into a 50 mL Teflon lined stainless steel autoclave and kept at 180 °C for 24 h. The resulted white precipitate was collected by centrifugation, washed repeatedly with ethanol and water, and dried at 70 °C for 12 h. The acquired powder was dispersed in 700 mL NaOH aqueous solution (0.1 mol/L), stirred for 24 h at RT, centrifuged, and washed repeatedly with water until the pH value of aqueous solution was of 7–8.

Synthesis of anatase $TiO_2\{100\}$ and $TiO_2\{101\}$ NCs: typically, 6.6 mL $TiCl_4$ was added dropwise into 20 mL HCl aqueous solution

(0.43 mol/L) at 0 °C. After stirring for an additional 0.5 h, the solution was added dropwise into 50 mL $NH_3$ aqueous solution (5.5 wt%) under stirring at RT. Then the pH value of the solution was adjusted to between 6 and 7 using 4 wt% $NH_3$ aqueous solution, after which the system was stirred at RT for 2 h. The resulted precipitate was filtered, washed repeatedly with water until no residual $Cl^-$ could be detected, and then dried at 70 °C for 12 h to acquire $Ti(OH)_4$ precursor. To prepare anatase $TiO_2$-{100} nanocrystals, 2.0 g $Ti(OH)_4$ and 0.5 g $(NH_4)_2SO_4$ were dispersed in a mixture of 15 mL $H_2O$ and 15 mL iso-propanol under stirring at RT, then the mixture was transferred into a 50 mL Teflon-lined stainless steel autoclave and kept at 180 °C for 24 h. The obtained white precipitate was collected and washed repeatedly with water. To prepare anatase $TiO_2$-{101} nanocrystals, 2.0 g $Ti(OH)_4$ and 0.2 g $NH_4Cl$ were dispersed in a mixture of 15 mL $H_2O$ and 15 mL isopropanol under stirring at RT, then the mixture was transferred into a 50 mL Teflon-lined stainless steel autoclave and kept at 180 °C for 24 h. The obtained white precipitate was collected and washed repeatedly with water.

Synthesis of anatase $TiO_2$ NCs-$C_3N_4$ composites: calculated amounts of dicyandiamide ($C_2H_4N_4$) and $TiO_2$ NCs were mixed in a crucible. The crucible was placed into a tube furnace, purged in Ar 1 h, and heated to 550 °C at a rate of 2.5 °C/min and kept for 4 h, then cooled to room temperature. The acquired powders were taken out and grind to obtain $TiO_2$ NCs-$C_3N_4$ composites.

## Structure characterizations

Powder X-ray diffraction (XRD) patterns were recorded on a Philips X'Pert Pro Super diffractometer with Cu Kα radiation ($\lambda = 0.15406$ nm) operated at 40 kV and 50 mA. Transmission infrared spectra were recorded on a Nicolet 8700 spectrometer at room temperature. Electron paramagnetic resonance (ESR) spectra with and without Xenon lamp irradiation were recorded on a JEOL JES-FA200 ESR spectrometer (9.063 GHz, X-band) at 130 K with employed microwave power, modulation frequency, and modulation amplitude of 0.998 mW, 100 kHz, and 0.35 mT, respectively. Steady-state photo-luminescence spectra were measured on a HORIBA LabRAM HR spectrograph with a continuous wave 325 nm laser as the exciting source and the signal was collected by passing through a filter with cut-off wavelengths below 380 nm. UV–vis diffuse reflectance spectra (UV–vis DRS) were obtained on a Shimadzu DUV-3700 spectro-photometer equipped with an integrating sphere attachment. X-ray photoelectron spectroscopy (XPS) measurements were performed on an ESCALAB 250 high-performance electron spectrometer using monochromatized Al Kα (hv = 1486.7 eV) as the excitation source, and the likely charging of samples was corrected by setting the C 1$s$ binding energy of the adventitious carbon to 284.8 eV. Near-edge X-ray absorption fine structure (NEXAFS) spectra were measured at photo-electron spectroscopy end-station of National Synchrotron Radiation Laboratory. Transmission electron microscopy (TEM), high-resolution transmission electron microscopy (HRTEM) and element mapping images were performed with a JEOL JEM-2100F instrument at an acceleration voltage of 120 kV.

Adsorption microcalorimetry measurements were carried out on a home-setup equipment consisting of a Setaram Sensys EVO 600 Tian-Calvet microcalorimeter and an Micromeritics Autochem II 2920 automated chemisorption apparatus[46]. Typically, 50 mg sample was placed in the sample quartz tube and degassed in He (flow rate: 50 mL/min) at 200 °C for 60 min, then the sample was cooled to −100 °C, and the gas stream was switched to 2% $CH_4$/He (flow rate: 50 mL/min) for adsorption. After $CH_4$ adsorption reached saturation, the gas stream was switched back to He (flow rate: 50 mL/min) for desorption. The adsorption/desorption amounts and accompanying heat flows were quantified by the chemisorption apparatus and micro-calorimeter, respectively, from which the adsorption/desorption heats were calculated.

In situ DRIFTS experiments were performed at 298 K on a Thermo Scientific Nicolet iS50 FTIR Spectrometer with a mercury cadmium telluride detector cooled with liquid nitrogen. The spec-trometer was equipped with a Harrik Praying Mantis diffuse reflec-tion accessory and a Harrick high-temperature reaction cell with ZnSe windows. The reaction cell was connected to a SH-110 dry scroll vacuum pump (Agilent Technologies), $H_2O_2$ aqueous solution stored in a quartz tube welded with Kovar, and 8%$CH_4$ + 4%$O_2$ + 88% Ar gas via three closed valves. The 30% $H_2O_2$ aqueous solution was purified by repeated cycles of freeze−pump−thaw treatments. The UV light irradiation on the sample was accomplished through the front window of the high-temperature reaction chamber using a 100 W high-pressure Hg arc lamp (Oriel 6281), which provides a pressure-broadened emission spectrum from gaseous Hg in the UV-light region. A water filter was used to remove the IR portion of the emission spectrum. Typically, the sample was loaded in the sample holder of the reaction cell, then the reaction cell was evacuated by opening the valve connecting the vacuum pump. After the pressure decreased to 10 Torr, the valve connecting the vacuum pump was closed, and the valve connecting the $H_2O_2$ aqueous solution was opened to reach a stable pressure, and then the valve connecting to 8%$CH_4$ + 4%$O_2$ + 88%Ar gas was open to allow the pressure of the reaction cell to 1 atm, and finally both valves were closed. The DRIFTS spectrum of the sample prior to UV light illumination was firstly taken as the background spectrum, then the UV light was turned on to irradiate the sample and the DRIFTS spectra were taken in a sequential mode. The DRIFTS spectrum of the sample was also taken after the turn off of the UV light. All DRIFTS spectra were measured with 128 scans at a resolution of 4 cm$^{-1}$.

## Photocatalytic activity measurements

Photocatalytic activity of various samples in aqueous-phase methane conversion was evaluated in a quartz reactor with a cooling-water jacket to maintain the reaction temperature at 25 °C under atmo-spheric pressure using a 300 W Xe lamp as the light source whose spectrum is shown in the Supplementary Fig. 35. Typically, 20 mg photocatalyst, 20 mL deionized water and a certain amount of $H_2O_2$ aqueous (1 mol/L) solution were mixed in the reactor. The reaction system was adequately deaerated by reaction gas for 1 h, and then was irradiated by the Xe lamp. then the photocatalytic reaction was carried out. After a desirable reaction time, 0.5 mL gas was sampled from the reaction system and its composition products was analyzed by a Fuli GC9720 gas chromatography equipped with FID and TCD detectors.

Liquid-phase oxygenate products were analyzed and quantified by $^1H$ nuclear magnetic resonance (NMR) spectra acquired on a JEOL ECS 400 MHz NMR spectrometer. A DSS solution in $D_2O$ (0.020 wt.%) with the $^1H$ chemical shift at $\delta = 0.0$ ppm was prepared to calibrate the chemical shift. Typically, 0.70 mL clear aqueous solution was sampled from the reaction system and mixed with 0.10 mL DSS solution in a NMR tube and the $^1H$ NMR spectrum was taken. The intensity of measured $^1H$ NMR peak of various products were compared to the corresponding $^1H$ NMR working curve acquired using pure product of different concentrations (Supplementary Fig. 36). Since pure $CH_3OOH$ could not be purchased while both $CH_3OOH$ and $CH_3OH$ have the methyl group, the amount of $CH_3OOH$ in the liquid-phase products was quantified using the working curve of $CH_3OH$[22].

The concentration of HCHO was quantified by the colorimetric method[22]. Typically, 100 mL of the reagent aqueous solution was prepared by dissolving 15 g ammonium acetate, 0.3 mL acetic acid, and 0.2 mL pentane-2,4-dione in water. Then, 0.5 mL liquid product was mixed with 2.0 mL water and 0.5 mL reagent solution. The mixed solution was maintained at 35 °C and measured by UV − vis absorption spectrum until the absorption intensity at 412 nm did not further

increase. The concentration of HCHO in the liquid product was determined by the standard curve (Supplementary Fig. 37).

The concentration of $H_2O_2$ in the aqueous solution was quantified by the colorimetric method[13,21]. Typically, a reagent aqueous solution was prepared by dissolving 0.636 g $K_2TiO(C_2O_4)_2$ and 20 μL concentrated $H_2SO_4$ (98%) in 100 mL deionized water. 0.2 mL aqueous solution was exacted from the reaction system and mixed with 4.0 mL reagent solution. Then the UV – vis absorption spectrum of the mixed solution was measured, and the intensity of the absorption peak at 398 nm arising from the complex formed by $K_2TiO(C_2O_4)_2$ and $H_2O_2$ was compared to the working curve acquired using pure $H_2O_2$ aqueous solution of different concentrations (Supplementary Fig. 38) to quantify the $H_2O_2$ concentration.

Methane conversion, product selectivity, $H_2O_2$ conversion and $H_2O_2$ utilization efficiency were calculated as the following:

$$\text{Methane conversion} (\%) = (n(CH_4)_{\text{before reaction}}$$
$$- n(CH_4)_{\text{after reaction}})/n(CH_4)_{\text{before reaction}} \times 100\%$$

$$\text{Product selectivity} (\%) = n_{\text{Product}}/(n(CH_4)_{\text{before reaction}} - n(CH_4)_{\text{after reaction}}) \times 100\%$$

$$H_2O_2 \text{conversion} (\%) = \left(n(H_2O_2)_{\text{before reaction}}\right.$$
$$\left. - n(H_2O_2)_{\text{after reaction}}\right)/n(H_2O_2)_{\text{before reaction}} \times 100\%$$

$$H_2O_2 \text{utilization efficiency} (\%) = \left(n(H_2O_2)_{\text{before reaction}}\right.$$
$$\left. - n(H_2O_2)_{\text{after reaction}} - n(H_2O_2)_{\text{decomposition to O2}}\right)/n(H_2O_2)_{\text{before reaction}} \times 100\%$$

$$n(H_2O_2)_{\text{decomposition to O2}} = n(O_2)_{\text{after reaction}} - n(O_2)_{\text{before reaction}} - n_{\text{O2reacted}}$$

$$\text{Carbon balance} (\%) = n_{\text{carbon in all products}}/$$
$$(n(CH_4)_{\text{before reaction}} - n(CH_4)_{\text{after reaction}}) \times 100\%$$

Where n was the quantified amount of reactants or products, while $n_{\text{O2 reacted}}$ was calculated from the amount of products and the ratio of the products formed by $O_2$ based on the isotope-labelling results. For photocatalytic reactions using $^{18}O_2$, $n_{18O2\,\text{reacted}}$ was calculated by $(n(^{18}O_2)_{\text{before reaction}}-n(^{18}O_2)_{\text{after reaction}})$, in which $n(^{18}O_2)$ was quantified using GC-MS. The carbon balance was calculated not less than 96.7% for all studied photocatalytic reactions.

### Product analysis of photocatalytic reactions using isotope-labelled reactants

Liquid-phase oxygenates produced by aqueous-phase photocatalytic methane conversion using $^{13}CH_4$ were analyzed by $^1H$ NMR and $^{13}C$ with decoupling NMR spectrometer as described above. Products of aqueous-phase photocatalytic methane conversion using $^{18}O_2$ and or $H_2^{18}O$ were analyzed by mass spectrometer as the following: 0.5 mL gas was sampled from the reaction system and its composition was analyzed on a Trace GC/ISQ MS; and 3 mL clear aqueous solution was sampled from the reaction system and transferred into a quartz tube welded with Kovar and then connected to a QIC20 mass spectrometer (Heiden Analytical Ltd.) and a Hicube 80 Eco pump station by two closed valves. The aqueous solution was purified by repeated cycles of freeze–pump–thaw treatments and its composition was analyzed by the QIC20 mass spectrometer.

### Theoretical calculations

All theoretical calculations were carried out using the Vienna ab initio simulation package (VASP)[47,48], and the exchange-correlation term was described by the Perdew, Burke and Ernzerhof version within the generalized gradient approximation (PBE-GGA)[49]. The project-augmented wave (PAW)[50,51] method was used to represent the core-valence electron interaction. The titanium 3 s, 3p, 3d, 4 s, and the carbon and oxygen 2 s, 2p electrons were treated as valence electrons and an energy cutoff of 400 eV for the basis-set expansion was used. The anatase $TiO_2(001)$-(1 × 4), $TiO_2(101)$ and $TiO_2(100)$ surface was modeled as a periodic slab with six O-Ti-O trilayers of oxide. A vacuum between slabs >15 Å and corresponding 1 × 1 × 1 k-point mesh were used during the calculations. Adsorption was modeled on one side of the slab, and during structural optimizations, all of the atoms except those in the bottom $TiO_2$ trilayer of the slab, were allowed to relax until atomic forces reached below 0.05 eV/Å. The adsorption energy ($E_{\text{ads}}$) was expressed using the average adsorption energy calculated by $E_{\text{ads}} = E_{\text{ad/sub}} - (E_{\text{sub}} + E_{\text{ad}})$ in which $E_{\text{ad/sub}}$ is the total energy of the interacting system containing adsorbed molecules and $TiO_2$ support in a surface cell, $E_{\text{sub}}$ is the total energy of the anatase $TiO_2$ slab and $E_{\text{ad}}$ is the total energy of the molecule in gas phase.

Details on structural characterizations, activity evaluations, and DFT calculations can be found in the supplementary information.

## Data availability

The data supporting the findings of the study are available within the paper and its Supplementary Information. Source data are provided with this paper.

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

## Acknowledgements

This work was financially supported by the National Natural Science Foundation of China (21525313, 91745202, 92145302), the Chinese Academy of Sciences, the Fundamental Research Funds for the Central Universities (20720220008), the Changjiang Scholars Program of Ministry of Education of China, and University of Science and Technology of China (KY2060000176).

## Author contributions

W.H. conceived and supervised the project, oversaw all data analysis and discussion. X.S. performed the experiments and analyzed the data. X.C. performed the calculations. C.F., Q.Y., and F.F. assisted with the experiments. X.Z. and J.Z. assisted with the NEXAFS measurements. Y.L. discussed the data. W.Z. supervised the calculations. W.H. and X.S. prepared the paper.

## Competing interests

The authors declare no competing interests.
