## [Peer Review File · Nature Communications]

Title: Molecular oxygen enhances H₂O₂ utilization for the photocatalytic conversion of methane to liquid-phase oxygenatesREVIEWER COMMENTS

Reviewer #1 (Remarks to the Author):

This research uses molecular oxygen to improve H₂O₂ utilization efficiency for photocatalytic CH₄ oxidation. The authors did provide many evidences to establish a plausible reaction mechanism. However, it is not enough to be accepted by NC, I suggest reject it. Several points have to be addressed to make it a complete work.

1 As photocatalyst, its photophysical and photoelectrical properties before and after synthesis should be studied, such as UV-vis diffuse reflectance spectra, band structures, photo-response curves, electrochemical impedance spectroscopies. The authors should give more discussions in main text.

2 In photocatalytic CH₄ conversion, are all products formed at the same time? Or in particular order? It is suggested to study the change of product concentration over reaction time.

3 How about the light and chemical stabilities of catalyst after being used many times? And catalytic effect?

4 The statement “These oxygen isotope labelling experimental results suggest that O₂ mainly acts as a promoter for photocatalytic CH₄ conversion with H₂O₂, instead of as a reactant” is less rigorous. (Page 5, Paragraph 1, Last line) The conducted oxygen isotope labeling experiments only seem to prove that the oxygen in liquid products mainly comes from H₂O₂, rather than oxygen and water, but it cannot be inferred that oxygen promotes photocatalytic CH₄ conversion.

5 Please add the decomposition rate of H₂O₂ into O₂ by catalysts in the presence of H₂O₂, catalyst and/or light.

Reviewer #2 (Remarks to the Author):

In this manuscript, the author reported that the introduction of O₂ could enhance the utilization efficiency of H₂O₂ by suppressing H₂O₂ adsorption on oxides and consequent photogenerated holes-mediated H₂O₂ dissociation into O₂. Detailed characterizations and DFT study were carried out to explain the effect of Z-scheme TiO₂-C₃N₄ heterojunctions to photocatalytic methane oxidation with H₂O₂. However, there are several crucial issues to be solved.

1. The lattice fringe should be re-measured and lattice spacing in TEM images must be calculated again because the yellow lines in TEM images were misplaced and ambiguous. The lattice spacing of C₃N₄ should be provided to explain the successful synthesis for heterojunction catalysts (Z-scheme TiO₂-C₃N₄).

2. The light wavelength of 300 W Xe lamp should be added. Blank experiments must be made if the

wavelength of light illumination was UV range, for example, no catalysts and no CH₄ with TiO₂{001}-C₃N₄ catalyst to exclude the carbon source on C₃N₄ support.

3. The authors only showed the selectivity of various oxygenates, but the yield or productivity of all products should be exhibited especially in the manuscript to compare the change of photocatalytic activity after adding O₂.

4. For products analysis, CH₂O species at 1712 cm⁻¹ were detected on in situ DRIFTS spectra, and ·CHO radicals were proposed in the reaction paths of TiO₂ NCs and TiO₂ NCs-C₃N₄. However, the authors did not quantify the generation of HCHO. According to enormous references on photocatalytic methane oxidation over TiO₂ photocatalysts, HCHO can be the prominent over-oxidized products. Therefore, it is necessary to provide analysis about HCHO.

5. We disagreed that “O₂ mainly acts as a promoter for photocatalytic CH₄ conversion with H₂O₂, instead of as a reactant” at line 154. Firstly, ·O₂-radicals were detected by in situ ESR spectra, it is easy to form ·OOH by combination ·O₂- with H⁺. Secondly, the authors presented O₂ participated in the generation of CH₃OOH in Figure 2. Finally, CH₃18OH, HCO18OH, CH₃HC218OH and CH₃CO18OH were detected by mass spectra when using 18O₂. So the above conclusions is contradictory with “O₂ mainly acts as a promoter for photocatalytic CH₄ conversion with H₂O₂, instead of as a reactant”.

6. If authors wanted to obtain the conclusion about “photocatalytic CH₄ activation to ·CH₃ radicals is mainly mediated by h⁺” in line 179, the “DMPO-CH₃” signals between the in situ ESR spectra of CH₄ and that of CH₄+isopropyl alcohol mixture in Figure S12 should be compared under the same level. The author should provide in situ ESR spectra of CH₄+hole trapping agents to determine the oxidation of methane to ·CH₃ is mainly by h⁺.

7. The author should recheck the assignments of vibrational bands observed in the in situ DRIFTS spectra especially about carbonates (1504 and 1592 cm⁻¹) in methane oxidation reaction. Meanwhile, in situ DRIFTS spectra among the 2100-2400 cm⁻¹ should be given to exclude the generation of CO_x.

8. The photocatalytic reaction path of CH₄ photocatalytic in Figure 2g and 2h was inadequate. For example, the generation of ·O₂- radicals should be represented on the schemes. The author proposed the formation of CH₃OH was mainly the reduction of CH₃OOH, but it was easy to form CH₃OH by the coupling of ·CH₃ and ·OH. In addition, authors did not explain the oxidation pathway of HCOOH from ·OH radicals or holes.

Reviewer #3 (Remarks to the Author):

Comments on the manuscript NCOMMS-22-18924-T entitled “Oxygen as a Molecular Additive to Enhance Utilization Efficiency of H₂O₂ for Photocatalytic Conversion of Methane to Liquid-Phase Oxygenates” by Weixin Huang et al.

This paper reported the photocatalytic methane (CH₄) conversion by H₂O₂ as the oxidant over TiO₂{001}-C₃N₄ composite photocatalysts, in which, interestingly, O₂ is used an molecular additive to enhance the utilization efficiency of H₂O₂ up to 93.3%, and the selectivity of formic acid and liquid-phase oxygenates is up to 69.8% and 97%, respectively. According to the studies of the NMR, ESR, and

in-situ FTIR techniques, the photocatalytic mechanism of CH₄ is proposed. Thus, I can recommend the publication of this manuscript after a major revision:

1. An interesting discovery that O₂ can enhance the utilization efficiency of H₂O₂ effectively was proposed, which, however, was not discussed comprehensively how O₂ can improve the utilization efficiency of H₂O₂. It was found that the oxygen atoms in CH₃OOH, CH₃OH, HCOOH and CH₃CH₂OH are contributed majorly by H₂O₂ and minor by O₂, but seldom by H₂O, while the oxygen atoms of CH₃COOH are contributed majorly by H₂O, and minor by O₂. (Figure 2) As such, the authors proposed that O₂ mainly acts as a promoter for photocatalytic CH₄ conversion with H₂O₂, instead of as a reactant. (Page 5, line 155) This expression is inappropriate, because O₂ is clearly involved in the reaction, and the oxygen atom of O₂ transfers into all the products. The low proportion of products from O₂ oxidation may be due to the low amount of O₂. Therefore, it is necessary to detect and discuss the activity and selectivity of photocatalytic CH₄ conversion by H₂O₂ with the variable amount of O₂.
2. According to the previous reports, HCHO should be formed in the photocatalytic reaction over TiO₂ photocatalysts. It should be discussed why the HCHO is not produced in the CH₄ oxidation herein.
3. The ¹³CH₄ was used to demonstrate that all products exclusively form from CH₄. However, if the C of the products is originated from CH₄, why are not the splitting peaks due to ¹H–¹³C J coupling (~140 Hz) for ¹³CH₃OH, ¹³CH₃¹³CH₂OH, and ¹³CH₃COOH observed in these ¹H NMR spectra (Figure S3)? Furthermore, all the ¹H NMR signals of the ¹³CH₄ and its oxidation products shift to high field in relation to the ¹H NMR signals of the natural CH₄, which should be because the magnetic field is unstable during the NMR experiment. Thus, it is necessary to redo the ¹H NMR experiments with stable magnetic field.
4. The ESR experiments were performed in the presence of DMPO at low temperature (130K), rather than under approximate reaction conditions. Thus, it should be ex-situ experiment, rather than in-situ experiment.

Author reply to Reviewer 1's comments

This research uses molecular oxygen to improve H₂O₂ utilization efficiency for photocatalytic CH₄ oxidation. The authors did provide many evidences to establish a plausible reaction mechanism. However, it is not enough to be accepted by NC, I suggest reject it. Several points have to be addressed to make it a complete work.

Author reply: We appreciate the reviewer's insightful comments very much. In our submission, we report for the first time O₂-suppressed H₂O₂ adsorption on oxides and consequent photogenerated holes-mediated H₂O₂ dissociation into O₂, which provides a promising new strategy to achieve high H₂O₂ utilization efficiency and excellent photocatalytic performance for photocatalytic oxidation reactions over oxide-based photocatalysts via co-use of H₂O₂ and O₂. In the photocatalytic aqueous-phase methane conversion over an optimized TiO₂{001}-C₃N₄ composite photocatalyst during our study, O₂ additive significantly enhances the utilization efficiency of H₂O₂ unprecedentedly up to 93.3%, giving formic acid and liquid-phase oxygenates selectivity respectively of 69.8% and 97% and formic acid yield of 486 μmol_{HCOOH}·g_{catalyst}⁻¹·h⁻¹. Since H₂O₂ production is well known as an environment-unfriendly and economic-costly process, in addition to excellent photocatalytic performance, high H₂O₂ utilization efficiency is also very desirable in photocatalytic selective conversion of methane over oxide-based photocatalysts widely using H₂O₂. Therefore, we believe that our results truly represent a breakthrough and will attract great attention and exert profound influences. Meanwhile, we have seriously considered the reviewer's comments and revised the manuscript accordingly. We hope that the revised manuscript will be suitable for the publication in *Nature Communications*.

1 As photocatalyst, its photophysical and photoelectrical properties before and after synthesis should be studied, such as UV-vis diffuse reflectance spectra, band structures, photo-response curves, electrochemical impedance spectroscopies. The authors should give more discussions in main text.

Author reply: We appreciate the reviewer's kind suggestions and accept them. We have added a paragraph to describe photophysical and photoelectrical properties of various TiO₂ NCs and TiO₂ NCs-C₃N₄ composite photocatalysts in the revised manuscript as the following:

“The band structures of various TiO₂ NCs and TiO₂ NCs-C₃N₄-0.1 photocatalysts were determined using UV-vis spectra and valence band XPS spectra (Supplementary Fig. 21). TiO₂ NCs-C₃N₄-0.1 exhibits smaller band gaps than corresponding the TiO₂ NCs, suggesting stronger capacity for light absorption and charge generation. The conduction band edges of TiO₂ NCs and TiO₂ NCs-C₃N₄ composites were measured to be -0.14~-0.34 and -0.41~-0.47 vs RHE, respectively, consistent with the experimental observations that H₂O₂ undergoes the e⁻-mediated activation into ·OH radicals over TiO₂ NCs and ·OOH radicals over TiO₂ NCs-C₃N₄-0.1 composites. ESR spectra (Supplementary Fig. 22a) show that TiO₂ NCs-C₃N₄-0.1 exhibit much lower densities of F⁺ and Ti³⁺ defects than TiO₂ NCs and the defect density follows an order of TiO₂{101} > TiO₂{100} > TiO₂{001} > TiO₂{101}-C₃N₄-0.1 > TiO₂{100}-C₃N₄-0.1 > TiO₂{001}-C₃N₄-0.1. Accordingly, PL spectra (Supplementary Fig. 22b) show that the PL peak arising from the recombination of photoexcited electrons and holes displays an intensity order of TiO₂{101} > TiO₂{100} > TiO₂{001} > TiO₂{101}-C₃N₄-0.1 > TiO₂{100}-C₃N₄-0.1 > TiO₂{001}-C₃N₄-0.1. EIS spectra of various TiO₂ NCs and TiO₂ NCs-C₃N₄-0.1 photocatalysts were also measured, in which a smaller radius

represents a low charge transfer resistance. All photocatalysts exhibit semicircle EIS spectra (Supplementary Fig. 22c), and the semicircle radius and consequently the charge transfer resistance follow an order of $\text{TiO}_2\{101\} > \text{TiO}_2\{100\} > \text{TiO}_2\{001\} > \text{TiO}_2\{101\}\text{-C}_3\text{N}_4\text{-0.1} > \text{TiO}_2\{100\}\text{-C}_3\text{N}_4\text{-0.1} > \text{TiO}_2\{001\}\text{-C}_3\text{N}_4\text{-0.1}$. ESR, PL and EIS are all bulk-sensitive characterization techniques, and their characterization results show that $\text{TiO}_2\text{ NCs-C}_3\text{N}_4\text{-0.1}$ exhibit higher charge separation and transfer efficiencies than corresponding $\text{TiO}_2\text{ NCs}$ and that $\text{TiO}_2\{001\}$ is the best of various $\text{TiO}_2\text{ NCs}$ while $\text{TiO}_2\{001\}\text{-C}_3\text{N}_4\text{-0.1}$ is the best of $\text{TiO}_2\text{ NCs-C}_3\text{N}_4\text{-0.1}$ composite photocatalysts, consistent with the photocatalytic activity results.” (Please see Lines 259-281).

2 In photocatalytic CH_4 conversion, are all products formed at the same time? Or in particular order? It is suggested to study the change of product concentration over reaction time.

Author reply: We appreciate the reviewer’s insightful comments very much. Following the reviewer’s comments, we examined initial evolutions of reaction products as a function of reaction time over $\text{TiO}_2\{001\}\text{-C}_3\text{N}_4\text{-0.1}$. The results have been added in the revised manuscript as the following:

“Initial evolutions of reaction products as a function of reaction time were examined over $\text{TiO}_2\{001\}\text{-C}_3\text{N}_4\text{-0.1}$ (Supplementary Table 11). At a reaction time of 10 min, CH_3OOH , CH_3OH and HCHO were detected, and CH_3OOH was the major product. The CH_3OOH , CH_3OH and HCHO productions increased at a reaction time of 30 min, meanwhile, HCOOH and $\text{CH}_3\text{CH}_2\text{OH}$ appeared. As a reaction time of 1 h, the CH_3OOH production decreased and HCHO was not detected, whereas the CH_3OH and HCOOH productions increased greatly and the $\text{CH}_3\text{CH}_2\text{OH}$ production increased slightly, meanwhile, CH_3COOH emerged. These observations suggest CH_3OOH as the primary product and CH_3OH , HCHO , HCOOH , $\text{CH}_3\text{CH}_2\text{OH}$ and CH_3COOH as the secondary products that are produced sequentially. Moreover, the reaction rate of HCHO seems to be faster than the formation rate.” (please see Lines 145-155)

3 How about the light and chemical stabilities of catalyst after being used many times? And catalytic effect?

Author reply: We appreciate the reviewer’s insightful comments very much. Following the reviewer’s comments, we evaluated the stability of $\text{TiO}_2\{001\}\text{-C}_3\text{N}_4\text{-0.1}$ photocatalyst and found that $\text{TiO}_2\{001\}\text{-C}_3\text{N}_4\text{-0.1}$ is stable and its performance maintains well within six cycles of photocatalytic activity evaluations. Routine structural characterization results, including XPS, XPS, UV-Vis spectra and photocurrent measurements, show few difference between the as-synthesized and used $\text{TiO}_2\{001\}\text{-C}_3\text{N}_4\text{-0.1}$ catalysts. by initial evolutions of reaction products as a function of reaction time over. The results have been added in the revised manuscript as the following:

“ $\text{TiO}_2\{001\}\text{-C}_3\text{N}_4\text{-0.1}$ is stable and its performance maintains well within six cycles of photocatalytic activity evaluations (Supplementary Fig. 4). Routine structural characterization results (Supplementary Fig. 5), including XPS, VB-XPS, UV-Vis spectra and photocurrent measurements, show few difference between the as-synthesized and used $\text{TiO}_2\{001\}\text{-C}_3\text{N}_4\text{-0.1}$ catalysts.” (please see Lines 135-139)

4 The statement “These oxygen isotope labelling experimental results suggest that O_2 mainly acts as a promoter for photocatalytic CH_4 conversion with H_2O_2 , instead of as a reactant” is less rigorous. (Page 5, Paragraph 1, Last line) The conducted oxygen isotope labeling experiments only seem to

prove that the oxygen in liquid products mainly comes from H₂O₂, rather than oxygen and water, but it cannot be inferred that oxygen promotes photocatalytic CH₄ conversion.

Author reply: We appreciate the reviewer's insightful comments very much. We have rewritten the commented sentence more rigorously in the revised manuscript as the following:

“Therefore, during photocatalytic aqueous-phase CH₄ conversion in the presence of H₂O₂ and O₂, CH₄ preferentially reacts with H₂O₂ to produce liquid-phase oxygenates, while O₂ acts mainly as a promoter to enhance H₂O₂ utilization efficiency and consequently CH₄ conversion, and minorly as a reactant.” (please see Lines 207-210)

5 Please add the decomposition rate of H₂O₂ into O₂ by catalysts in the presence of H₂O₂, catalyst and/or light.

Author reply: We appreciate the reviewer's kind suggestions very much. We have added the decomposition rate of H₂O₂ under different conditions in Fig. 1e and Supplementary Tables 2 and 3, and described the results in the revised manuscript as the following:

“As shown in Fig. 1e, the H₂O₂ decomposition percentage/H₂O₂ decomposition rate/O₂ selectivity are 31.2%/610.9 μmol·h⁻¹/93.0% over TiO₂{001} NCs in the Ar atmosphere and decrease to 15.4%/301.5 μmol·h⁻¹/91.8% in the 10% O₂/Ar atmosphere, while they are 20.4%/399.4 μmol·h⁻¹/89.0% over TiO₂{001}-C₃N₄-0.1 in the Ar atmosphere and decrease to 8.26%/161.7 μmol·h⁻¹/86.4% in the 10% O₂/Ar atmosphere.” (please see Lines 88-93)

We appreciate the reviewer's effort on reviewing our manuscript very much, and we hope that the revised manuscript will be suitable for the publication in *Nature Communications*.

Author reply to Reviewer 2's comments

In this manuscript, the author reported that the introduction of O₂ could enhance the utilization efficiency of H₂O₂ by suppressing H₂O₂ adsorption on oxides and consequent photogenerated holes-mediated H₂O₂ dissociation into O₂. Detailed characterizations and DFT study were carried out to explain the effect of Z-scheme TiO₂-C₃N₄ heterojunctions to photocatalytic methane oxidation with H₂O₂. However, there are several crucial issues to be solved.

Author reply: We appreciate the reviewer's positive recommendation and insightful comments very much. We have seriously considered the reviewer's comments and revised the manuscript accordingly. We hope that the revised manuscript will be suitable for the publication in *Nature Communications*.

1. The lattice fringe should be re-measured and lattice spacing in TEM images must be calculated again because the yellow lines in TEM images were misplaced and ambiguous. The lattice spacing of C₃N₄ should be provided to explain the successful synthesis for heterojunction catalysts (Z-scheme TiO₂-C₃N₄).

Author reply: We appreciate the reviewer's insightful comments very much. We have re-measured lattice fringes and clearly marked the values and assigned crystal planes in HRTEM images shown in Supplementary Figs. 1 and 3. We are sure that the lattice fringes of 0.24 and 0.35 nm arise from anatase TiO₂ {001} and {101} crystal planes, respectively. We also re-took the HRTEM and elementary mapping images of our samples with higher qualities. Unfortunately, we failed to observe clear lattice fringes of g-C₃N₄ in the HRTEM images (Supplementary Fig. 3d), which is rather well known due to the strong damage effect of high-energy electron beam on the structure of g-C₃N₄, but its presence in the TiO₂ NCs-C₃N₄ composites is identified by XRD patterns (Supplementary Fig. 3e) and XPS spectra (Supplementary Fig. 3f).

In the revised manuscript, we have included a separate paragraph to describe synthesis and routine spectroscopic and microscopic characterization results of our TiO₂ NCs and TiO₂ NCs-C₃N₄ composite photocatalysts as the following:

Synthesis and structural characterizations. Anatase TiO₂ nanocrystals (NCs) predominantly enclosed by the {001} facets (denoted as TiO₂{001}), the {100} facets (denoted as TiO₂{100}) and the {101} facets (denoted as TiO₂{101}) were prepared following well-established recipes²⁷. XRD patterns, TEM and HRTEM images of as-synthesized various TiO₂ NCs (Fig. 1a, Supplementary Fig. 1) agree with those reported previously²⁷. TiO₂ NCs-C₃N₄ composites were prepared by calcination of mixture of calculated amounts of dicyandiamide (C₂H₄N₄) and TiO₂ NCs in Ar at 550 °C and denoted as TiO₂ NCs-C₃N₄-x, in which x was the actual TiO₂:C₃N₄ mole ratio acquired by TGA analysis (Supplementary Fig. 2 and Table 1). TEM, HRTEM and element mapping images (Figs. 1 b-d, Supplementary Fig. 3 a-c) show that various TiO₂ NCs preserve their original morphologies and form smooth anatase TiO₂-g-C₃N₄ interfaces. We failed to observe clear lattice fringes of g-C₃N₄ in the HRTEM images (Supplementary Fig. 3d) likely due to the strong damage effect of high-energy electron beam on the structure of g-C₃N₄, but its presence in the TiO₂ NCs-C₃N₄ composites is identified by XRD patterns (Supplementary Fig. 3e) and XPS spectra (Supplementary Fig. 3f).” (please see Lines 62-75)

2. The light wavelength of 300 W Xe lamp should be added. Blank experiments must be made if the

wavelength of light illumination was UV range, for example, no catalysts and no CH₄ with TiO₂{001}-C₃N₄ catalyst to exclude the carbon source on C₃N₄ support.

Author reply: We appreciate the reviewer's kind suggestions very much. We have added the spectrum of used Xe light in the revised Supplementary Information (please see Page S3).

We also did blank photocatalytic experiment of photocatalytic reaction in the presence of TiO₂{001}-C₃N₄-0.1 but absence of CH₄ in the reactant and did not detected any C-contained products. The results, together with the ¹³CH₄ experimental results, demonstrate that all C-contained products exclusively form from CH₄. We have described these results in the revised manuscript as the following:

“Blank photocatalytic experiment of photocatalytic reaction in the presence of TiO₂{001}-C₃N₄-0.1 but absence of CH₄ in the reactant did not produce detectable C-contained products; meanwhile, using ¹³CH₄, all C-contained products only contained ¹³C (Supplementary Fig. 6). Thus, all C-contained products exclusively form from CH₄.” (please see Lines 142-145)

3. The authors only showed the selectivity of various oxygenates, but the yield or productivity of all products should be exhibited especially in the manuscript to compare the change of photocatalytic activity after adding O₂.

Author reply: We appreciate the reviewer's kind suggestions very much. We have added the yields of all products in Fig. 1 f and g and Supplementary Tables 5-10, and described the results in the revised manuscript as the following:

“the HCOOH yield from 12.0 to 37.6 μmol·g_{catalyst}⁻¹·h⁻¹,” (please see Line 118)

“the HCOOH yield from 202.2 to 486 μmol·g_{catalyst}⁻¹·h⁻¹,” (please see Lines 125 and 126)

4. For products analysis, CH₂O species at 1712 cm⁻¹ were detected on in situ DRIFTS spectra, and ·CHO radicals were proposed in the reaction paths of TiO₂ NCs and TiO₂ NCs-C₃N₄. However, the authors did not quantify the generation of HCHO. According to enormous references on photocatalytic methane oxidation over TiO₂ photocatalysts, HCHO can be the prominent over-oxidized products. Therefore, it is necessary to provide analysis about HCHO.

Author reply: We appreciate the reviewer's insightful comments very much. Our in situ DRIFTS spectra demonstrate the formation of adsorbed HCHO as a surface intermediate during photocatalytic aqueous-phase CH₄ conversion to HCOOH using H₂O₂ and O₂. We analyzed all likely products but did not detect the formation of HCHO under the studied reaction conditions. As suggested by the reviewer 1, we examined initial evolutions of reaction products as a function of reaction time over TiO₂{001}-C₃N₄-0.1. The HCHO formation appears at a reaction time of 10 min, increases at a reaction time of 30 min, but disappears at a reaction time of 1 h. These observations suggest that the reaction rate of HCHO seems to be faster than the formation rate, which makes HCHO undetectable. We have added these results in the revised manuscript as the following:

“Initial evolutions of reaction products as a function of reaction time were examined over TiO₂{001}-C₃N₄-0.1 (Supplementary Table 11). At a reaction time of 10 min, CH₃OOH, CH₃OH and HCHO were detected, and CH₃OOH was the major product. The CH₃OOH, CH₃OH and HCHO productions increased at a reaction time of 30 min, meanwhile, HCOOH and CH₃CH₂OH appeared.

As a reaction time of 1 h, the CH₃OOH production decreased and HCHO was not detected, whereas the CH₃OH and HCOOH productions increased greatly and the CH₃CH₂OH production increased slightly, meanwhile, CH₃COOH emerged. These observations suggest CH₃OOH as the primary product and CH₃OH, HCHO, HCOOH, CH₃CH₂OH and CH₃COOH as the secondary products that are produced sequentially. Moreover, the reaction rate of HCHO seems to be faster than the formation rate.” (please see Lines 145-155)

5. We disagreed that “O₂ mainly acts as a promoter for photocatalytic CH₄ conversion with H₂O₂, instead of as a reactant” at line 154. Firstly, ·O₂⁻ radicals were detected by in situ ESR spectra, it is easy to form ·OOH by combination ·O₂⁻ with H⁺. Secondly, the authors presented O₂ participated in the generation of CH₃OOH in Figure 2. Finally, CH₃¹⁸OH, HCO¹⁸OH, CH₃HC₂¹⁸OH and CH₃CO¹⁸OH were detected by mass spectra when using ¹⁸O₂. So the above conclusions is contradictory with “O₂ mainly acts as a promoter for photocatalytic CH₄ conversion with H₂O₂, instead of as a reactant”.

Author reply: We appreciate the reviewer’s insightful comments very much. We have rewritten the commented sentence more rigorously in the revised manuscript as the following:

“Therefore, during photocatalytic aqueous-phase CH₄ conversion in the presence of H₂O₂ and O₂, CH₄ preferentially reacts with H₂O₂ to produce liquid-phase oxygenates, while O₂ acts mainly as a promoter to enhance H₂O₂ utilization efficiency and consequently CH₄ conversion, and minorly as a reactant.” (please see Lines 207-210)

6. If authors wanted to obtain the conclusion about “photocatalytic CH₄ activation to ·CH₃ radicals is mainly mediated by h⁺” in line 179, the “DMPO-CH₃” signals between the in situ ESR spectra of CH₄ and that of CH₄+isopropyl alcohol mixture in Figure S12 should be compared under the same level. The author should provide in situ ESR spectra of CH₄+hole trapping agents to determine the oxidation of methane to ·CH₃ is mainly by h⁺.

Author reply: We appreciate the reviewer’s insightful comments very much. We have compared the in situ ESR spectra of CH₄+H₂O and CH₄+ isopropyl alcohol+H₂O under UV light illumination in the presence of DMPO over TiO₂ NCs and TiO₂ NCs-C₃N₄-0.1 composites at 298 K in Supplementary Fig. 17 of revised manuscript. ·CH₃ radicals greatly grew when isopropanol was added to quench ·OH radicals, supporting that the formation of ·CH₃ radicals is not related to ·OH radicals. This also suggests the likely reaction between co-existing ·CH₃ and ·OH radicals. Meanwhile, ·CH₃ radicals were not observed in the in situ ESR spectra of CH₄+methanol mixture (CH₄ +5 mL CH₃OH+3μL DMPO) + H₂O₂ + O₂ under UV light illumination in the presence of DMPO over TiO₂ NCs and TiO₂ NCs-C₃N₄-0.1 composites at 298 K (Supplementary Fig. 18) due to the quench of h⁺ by methanol, but ·O₂⁻ radicals appear. Thus, photocatalytic CH₄ activation to ·CH₃ radicals is mainly mediated by h⁺, instead of by ·OOH, ·OH and ·O₂⁻ radicals.

The relevant results are described in the revised manuscript as the following:

“When CH₄ was introduced to the aqueous solutions containing TiO₂ NCs or TiO₂ NCs-C₃N₄-0.1 composites under UV light illumination (Fig. 2f), ·CH₃ radicals^{22,26}, in addition to ·OH radicals, were detected. They greatly grew when isopropanol was added to quench ·OH radicals (Supplementary Fig. 17), but could not be detected in the presence of H₂O₂ and O₂ when h⁺ was quenched using methanol (Supplementary Fig. 18). Thus, photocatalytic CH₄ activation to ·CH₃

radicals is mainly mediated by h^+ , instead of by $\cdot\text{OOH}$, $\cdot\text{OH}$ and $\cdot\text{O}_2^-$ radicals.” (please see Lines 227-233)

7. The author should recheck the assignments of vibrational bands observed in the in situ DRIFTS spectra especially about carbonates (1504 and 1592 cm^{-1}) in methane oxidation reaction. Meanwhile, in situ DRIFTS spectra among the 2100 - 2400 cm^{-1} should be given to exclude the generation of CO_x .

Author reply: We appreciate the reviewer’s insightful comments very much. As discussed in our manuscript, the observed vibrational bands (Supplementary Table 14) were assigned based on in situ DRIFTS spectra of CH_3OH and HCOOH adsorption on various TiO_2 NCs (Supplementary Fig. 28) and previous reports³⁴⁻³⁶ (please see Lines 355-358). The assignments of carbonates on TiO_2 have been ambiguous due to the presence of different types of carbonates. According to our own work [Ref. 34, *J. Phys. Chem. C* **120**, 21472–21485 (2016) and *ACS Catal.* **12**, 6457-6463 (2022)], the peak at 1592 cm^{-1} can be assigned to bidentate carbonate while that at 1504 cm^{-1} can be assigned to monodentate carbonate, We believe that our assignments are reliable.

In the revised manuscript, we have added the papers *J. Phys. Chem. C* **120**, 21472–21485 (2016) and *ACS Catal.* **12**, 6457-6463 (2022) as Ref. 35 and 36 to further support our assignments. We have also re-ordered all references accordingly.

Meanwhile, we have included the in situ DRIFTS spectra among the 2100 - 2400 cm^{-1} to Figure 4a in the revised manuscript, in which no CO_x signal could be seen.

8. The photocatalytic reaction path of CH_4 photocatalytic in Figure 2g and 2h was inadequate. For example, the generation of $\cdot\text{O}_2^-$ radicals should be represented on the schemes. The author proposed the formation of CH_3OH was mainly the reduction of CH_3OOH , but it was easy to form CH_3OH by the coupling of $\cdot\text{CH}_3$ and $\cdot\text{OH}$. In addition, authors did not explain the oxidation pathway of HCOOH from $\cdot\text{OH}$ radicals or holes.

Author reply: We appreciate the reviewer’s insightful comments very much. It is unlikely to schematically show the whole reaction network of our photocatalytic reactions in Fig. 2 g and h due to the complexity. The contents of Fig. 2 g and h are to schematically show the proposed dominant photocatalytic aqueous-phase CH_4 reaction paths to liquid-phase oxygenates in the presence of H_2O_2 and O_2 over TiO_2 NCs and TiO_2 NCs- C_3N_4 based on the photocatalytic reaction data, ESR results, isotope-labelling results, in situ DRIFTS results, and the literature’s reports. The description and discussion of Fig. 2 g and h can be found at Lines 231-255, 259-263, 362-365.

In the revised manuscript, we have re-written the caption of Fig. 2 g and h to clarify their contents as the following:

“Schematic diagrams of proposed dominant photocatalytic aqueous-phase CH_4 reaction paths to liquid-phase oxygenates in the presence of H_2O_2 and O_2 over (g) TiO_2 NCs and (h) TiO_2 NCs- C_3N_4 .” (please see Lines 169 and 170)

We appreciate the reviewer’s effort on reviewing our manuscript very much, and we hope that the revised manuscript will be suitable for the publication in *Nature Communications*.

Author reply to Reviewer 3's comments

This paper reported the photocatalytic methane (CH_4) conversion by H_2O_2 as the oxidant over $\text{TiO}_2\{001\}$ - C_3N_4 composite photocatalysts, in which, interestingly, O_2 is used a molecular additive to enhance the utilization efficiency of H_2O_2 up to 93.3%, and the selectivity of formic acid and liquid-phase oxygenates is up to 69.8% and 97%, respectively. According to the studies of the NMR, ESR, and in-situ FTIR techniques, the photocatalytic mechanism of CH_4 is proposed. Thus, I can recommend the publication of this manuscript after a major revision:

Author reply: We appreciate the reviewer's positive recommendation and insightful comments very much. We have seriously considered the reviewer's comments and revised the manuscript accordingly. We hope that the revised manuscript will be suitable for the publication in *Nature Communications*.

1. An interesting discovery that O_2 can enhance the utilization efficiency of H_2O_2 effectively was proposed, which, however, was not discussed comprehensively how O_2 can improve the utilization efficiency of H_2O_2 . It was found that the oxygen atoms in CH_3OOH , CH_3OH , HCOOH and $\text{CH}_3\text{CH}_2\text{OH}$ are contributed majorly by H_2O_2 and minor by O_2 , but seldom by H_2O , while the oxygen atoms of CH_3COOH are contributed majorly by H_2O , and minor by O_2 . (Figure 2) As such, the authors proposed that O_2 mainly acts as a promoter for photocatalytic CH_4 conversion with H_2O_2 , instead of as a reactant. (Page 5, line 155) This expression is inappropriate, because O_2 is clearly involved in the reaction, and the oxygen atom of O_2 transfers into all the products. The low proportion of products from O_2 oxidation may be due to the low amount of O_2 . Therefore, it is necessary to detect and discuss the activity and selectivity of photocatalytic CH_4 conversion by H_2O_2 with the variable amount of O_2 .

Author reply: We appreciate the reviewer's insightful comments very much. Following the reviewer's comments, we the O_2 concentration in the reactant was increased from 4% ($8\% \text{CH}_4 + 4\% \text{O}_2 + 88\% \text{Ar} + 165 \mu\text{L H}_2\text{O}_2 + 20\text{mL H}_2\text{O}$) to 12% ($8\% \text{CH}_4 + 12\% \text{O}_2 + 80\% \text{Ar} + 165 \mu\text{L H}_2\text{O}_2 + 20\text{mL H}_2\text{O}$), and the photocatalytic reaction was studied over $\text{TiO}_2\{001\}$ - C_3N_4 -0.1 comparatively with $^{16}\text{O}_2$ or $^{18}\text{O}_2$ in order to further clarify the role of O_2 . The results show that using $^{16}\text{O}_2$ or $^{18}\text{O}_2$ gave similar H_2O_2 utilization efficiencies of around 94% and slightly different CH_4 conversion rates and product selectivity. Using $^{18}\text{O}_2$, the $\text{CH}_3^{18}\text{OH}/\text{CH}_3^{16}\text{OH}$, $\text{HC}^{16}\text{O}^{18}\text{OH}/\text{HC}^{16}\text{O}^{16}\text{OH}$ and $\text{CH}_3\text{CH}_2^{18}\text{OH}/\text{CH}_3\text{CH}_2^{16}\text{OH}$ ratios in the liquid-phase products were measured respectively as around 0.19, 0.17 and 0.22, similar to the case of the reactant with 4% O_2 ; however, C^{18}O and C^{18}O_2 were detected and the fraction of $\text{C}^{16}\text{O}^{18}\text{O}$ in CO_2 is much larger than that of $\text{C}^{16}\text{O}^{16}\text{O}$, different from the case of the reactant with 4% O_2 . These results prove that, during photocatalytic aqueous-phase CH_4 conversion in the presence of H_2O_2 and O_2 , CH_4 preferentially reacts with H_2O_2 to produce liquid-phase oxygenates, while O_2 acts mainly as a promoter to enhance H_2O_2 utilization efficiency and consequently CH_4 conversion, and minorly as a reactant.

We have included these results in the revised manuscript as the following:

“In order to further clarify the role of O_2 , the O_2 concentration in the reactant was increased from 4% ($8\% \text{CH}_4 + 4\% \text{O}_2 + 88\% \text{Ar} + 165 \mu\text{L H}_2\text{O}_2 + 20\text{mL H}_2\text{O}$) to 12% ($8\% \text{CH}_4 + 12\% \text{O}_2 + 80\% \text{Ar} + 165 \mu\text{L H}_2\text{O}_2 + 20\text{mL H}_2\text{O}$), and the photocatalytic reaction was studied over $\text{TiO}_2\{001\}$ - C_3N_4 -0.1 comparatively with $^{16}\text{O}_2$ or $^{18}\text{O}_2$. Using $^{16}\text{O}_2$ or $^{18}\text{O}_2$ gave similar H_2O_2 utilization efficiencies of

around 94% and slightly different CH₄ conversion rates and product selectivity (Supplementary Table 13). Using ¹⁸O₂, the CH₃¹⁸OH/CH₃¹⁶OH, HC¹⁶O¹⁸OH/HC¹⁶O¹⁶OH and CH₃CH₂¹⁸OH/CH₃CH₂¹⁶OH ratios in the liquid-phase products were measured respectively as around 0.19, 0.17 and 0.22 (Supplementary Figs. 12-14), similar to the case of the reactant with 4% O₂; however, C¹⁸O and C¹⁸O₂ were detected and the fraction of C¹⁶O¹⁸O in CO₂ is much larger than that of C¹⁶O¹⁶O, different from the case of the reactant with 4% O₂. Therefore, during photocatalytic aqueous-phase CH₄ conversion in the presence of H₂O₂ and O₂, CH₄ preferentially reacts with H₂O₂ to produce liquid-phase oxygenates, while O₂ acts mainly as a promoter to enhance H₂O₂ utilization efficiency and consequently CH₄ conversion, and minorly as a reactant.” (please see Lines 197-210)

2. According to the previous reports, HCHO should be formed in the photocatalytic reaction over TiO₂ photocatalysts. It should be discussed why the HCHO is not produced in the CH₄ oxidation herein.

Author reply: We appreciate the reviewer’s insightful comments very much. We analyzed all likely products but did not detect the formation of HCHO under the studied reaction conditions. As suggested by the reviewer 1, we examined initial evolutions of reaction products as a function of reaction time over TiO₂{001}-C₃N₄-0.1. The HCHO formation appears at a reaction time of 10 min, increases at a reaction time of 30 min, but disappears at a reaction time of 1 h. These observations suggest that the reaction rate of HCHO seems to be faster than the formation rate, which makes HCHO undetectable. We have added these results in the revised manuscript as the following:

“Initial evolutions of reaction products as a function of reaction time were examined over TiO₂{001}-C₃N₄-0.1 (Supplementary Table 11). At a reaction time of 10 min, CH₃OOH, CH₃OH and HCHO were detected, and CH₃OOH was the major product. The CH₃OOH, CH₃OH and HCHO productions increased at a reaction time of 30 min, meanwhile, HCOOH and CH₃CH₂OH appeared. As a reaction time of 1 h, the CH₃OOH production decreased and HCHO was not detected, whereas the CH₃OH and HCOOH productions increased greatly and the CH₃CH₂OH production increased slightly, meanwhile, CH₃COOH emerged. These observations suggest CH₃OOH as the primary product and CH₃OH, HCHO, HCOOH, CH₃CH₂OH and CH₃COOH as the secondary products that are produced sequentially. Moreover, the reaction rate of HCHO seems to be faster than the formation rate.” (please see Lines 145-155)

3. The ¹³CH₄ was used to demonstrate that all products exclusively form from CH₄. However, if the C of the products is originated from CH₄, why are not the splitting peaks due to ¹H-¹³C J coupling (~140 Hz) for ¹³CH₃OH, ¹³CH₃¹³CH₂OH, and ¹³CH₃COOH observed in these ¹H NMR spectra (Figure S3)? Furthermore, all the ¹H NMR signals of the ¹³CH₄ and its oxidation products shift to high field in relation to the ¹H NMR signals of the natural CH₄, which should be because the magnetic field is unstable during the NMR experiment. Thus, it is necessary to redo the ¹H NMR experiments with stable magnetic field.

Author reply: We appreciate the reviewer’s insightful comments very much. We have inquired our colleague in charge of the NMR facility on which our ¹H NMR spectra were acquired, an expert on NMR. We were told that the frequency of the used NMR facility (Nuclear magnetic resonance spectrometer, AVANCE III HD400) is ~400 Hz, which is not able to induce the ¹H-¹³C J coupling (~140 Hz). Meanwhile, we measured the mass spectra of the products using ¹³CH₄, whose results only show ¹³C-contained products, furthering confirming that all products are from CH₄.

In the revised manuscript, the mass spectra of the products using $^{13}\text{CH}_4$ are added to Supplementary Fig. 6.

4. The ESR experiments were performed in the presence of DMPO at low temperature (130 K), rather than under approximate reaction conditions. Thus, it should be ex-situ experiment, rather than in-situ experiment.

Author reply: We appreciate the reviewer's insightful comments very much. All ESR spectra in the presence of DMPO were measured at 298 K under UV light illumination, and thus can be referred as in situ ESR. We have added the temperature to the figure captions in the revised manuscript as the following:

“(e) In situ ESR spectra of H_2O , $\text{H}_2\text{O}+\text{O}_2$, $\text{H}_2\text{O}+\text{H}_2\text{O}_2$ and $\text{H}_2\text{O}+\text{O}_2+\text{H}_2\text{O}_2$ solutions under UV light illumination at 298 K in the presence of DMPO over $\text{TiO}_2\{001\}$ NCs and $\text{TiO}_2\{001\}-\text{C}_3\text{N}_4-0.1$. (f) In situ ESR spectra of $\text{CH}_4+\text{H}_2\text{O}$ mixture under UV light illumination at 298 K in the presence of DMPO over $\text{TiO}_2\{101\}$, $\text{TiO}_2\{100\}$ and $\text{TiO}_2\{001\}$ NCs, $\text{TiO}_2\{101\}-\text{C}_3\text{N}_4-0.1$, $\text{TiO}_2\{100\}-\text{C}_3\text{N}_4-0.1$ and $\text{TiO}_2\{001\}-\text{C}_3\text{N}_4-0.1$ composites.” (please see Lines 165-169)

We appreciate the reviewer's effort on reviewing our manuscript very much, and we hope that the revised manuscript will be suitable for the publication in *Nature Communications*.

REVIEWER COMMENTS

Reviewer #1 (Remarks to the Author):

The authors answered all the issues from reviewers, it can be accepted by NC.

Reviewer #2 (Remarks to the Author):

The concerns have been well addressed in the revised manuscript. I recommend the publication of this manuscript in Nature Communications.

Reviewer #3 (Remarks to the Author):

Reviewer 3's comments 3: The $^{13}\text{CH}_4$ was used to demonstrate that all products exclusively form from CH_4 . However, if the C of the products is originated from CH_4 , why are not the splitting peaks due to $1\text{H}-^{13}\text{C}$ J coupling (~ 140 Hz) for $^{13}\text{CH}_3\text{OH}$, $^{13}\text{CH}_3^{13}\text{CH}_2\text{OH}$, and $^{13}\text{CH}_3\text{COOH}$ observed in these ^1H NMR spectra (Figure S3)? Furthermore, all the ^1H NMR signals of the $^{13}\text{CH}_4$ and its oxidation products shift to high field in relation to the ^1H NMR signals of the natural CH_4 , which should be because the magnetic field is unstable during the NMR experiment. Thus, it is necessary to redo the ^1H NMR experiments with stable magnetic field.

Author reply: We appreciate the reviewer's insightful comments very much. We have inquired our colleague in charge of the NMR facility on which our ^1H NMR spectra were acquired, an expert on NMR. We were told that the frequency of the used NMR facility (Nuclear magnetic resonance spectrometer, AVANCE III HD400) is ~ 400 Hz, which is not able to induce the $1\text{H}-^{13}\text{C}$ J coupling (~ 140 Hz). Meanwhile, we measured the mass spectra of the products using $^{13}\text{CH}_4$, whose results only show ^{13}C -contained products, further confirming that all products are from CH_4 .

According to the reply for the question 3 of Reviewer 3's comments, it is reasonable to believe that the author did not realize the seriousness of the question. Since the catalysts used by the author contain carbon, the sufficient evidence should be provided to prove that the product (such as CH_3OH , $\text{CH}_3\text{CH}_2\text{OH}$, CH_3COOH , and HCOOH) originates from the reactant (CH_4) rather than the organic substance contained in the catalyst. In the manuscript, the CH_4 was labeled by ^{13}C , and then the ^1H NMR technique was used to study the products of CH_4 oxidation. If the C of the products is originated from the $^{13}\text{CH}_4$, the splitting peaks due to $1\text{H}-^{13}\text{C}$ J coupling (~ 140 Hz) for $^{13}\text{CH}_3\text{OH}$, $^{13}\text{CH}_3^{13}\text{CH}_2\text{OH}$, $^{13}\text{CH}_3\text{COOH}$, and HCOOH should be observed definitely in the ^1H NMR spectra. However, the splitting peaks of these products do not occur, which proves that the product does not originate from the $^{13}\text{CH}_4$. To reply the comments, the authors inquired an expert on NMR, and were told that "the frequency of the used NMR facility (Nuclear magnetic resonance spectrometer, AVANCE III HD400) is ~ 400 Hz, which is not able to induce the $1\text{H}-^{13}\text{C}$ J coupling (~ 140 Hz)". We doubt the professionalism of this expert.

First, the magnetic field of the nuclear magnetic resonance spectrometer (AVANCE III HD400) should be ~400 MHz, rather than ~400 Hz. Second, the ~140 Hz of ^1H - ^{13}C J coupling corresponding to ~0.35 ppm can occur definitely in the spectra. Additionally, the authors do not explain why all the ^1H NMR signals of the $^{13}\text{CH}_4$ and its oxidation products shift to high field in relation to the ^1H NMR signals of the natural CH_4 .

Author reply to Reviewer 3' s comments

Reviewer 3' s comments 3: The $^{13}\text{CH}_4$ was used to demonstrate that all products exclusively form from CH_4 . However, if the C of the products is originated from CH_4 , why are not the splitting peaks due to $^1\text{H}-^{13}\text{C}$ J coupling (~ 140 Hz) for $^{13}\text{CH}_3\text{OH}$, $^{13}\text{CH}_3^{13}\text{CH}_2\text{OH}$, and $^{13}\text{CH}_3\text{COOH}$ observed in these ^1H NMR spectra (Figure S3)? Furthermore, all the ^1H NMR signals of the $^{13}\text{CH}_4$ and its oxidation products shift to high field in relation to the ^1H NMR signals of the natural CH_4 , which should be because the magnetic field is unstable during the NMR experiment. Thus, it is necessary to redo the ^1H NMR experiments with stable magnetic field.

Author reply: We appreciate the reviewer' s insightful comments very much. We have inquired our colleague in charge of the NMR facility on which our ^1H NMR spectra were acquired, an expert on NMR. We were told that the frequency of the used NMR facility (Nuclear magnetic resonance spectrometer, AVAVCE III HD400) is ~ 400 Hz, which is not able to induce the $^1\text{H}-^{13}\text{C}$ J coupling (~ 140 Hz). Meanwhile, we measured the mass spectra of the products using $^{13}\text{CH}_4$, whose results only show ^{13}C -contained products, furthering confirming that all products are from CH_4 .

According to the reply for the question 3 of Reviewer 3' s comments, it is reasonable to believe that the author did not realize the seriousness of the question. Since the catalysts used by the author contain carbon, the sufficient evidence should be provided to prove that the product (such as CH_3OH , $\text{CH}_3\text{CH}_2\text{OH}$, CH_3COOH , and HCOOH) originates from the reactant (CH_4) rather than the organic substance contained in the catalyst. In the manuscript, the CH_4 was labeled by ^{13}C , and then the ^1H NMR technique was used to study the products of CH_4 oxidation. If the C of the products is originated from the $^{13}\text{CH}_4$, the splitting peaks due to $^1\text{H}-^{13}\text{C}$ J coupling (~ 140 Hz) for $^{13}\text{CH}_3\text{OH}$, $^{13}\text{CH}_3^{13}\text{CH}_2\text{OH}$, $^{13}\text{CH}_3\text{COOH}$, and HCOOH should be observed definitely in the ^1H NMR spectra. However, the splitting peaks of these products do not occur, which proves that the product does not originate from the $^{13}\text{CH}_4$. To reply the comments, the authors inquired an expert on NMR, and were told that "the frequency of the used NMR facility (Nuclear magnetic resonance spectrometer, AVAVCE III HD400) is ~ 400 Hz, which is not able to induce the $^1\text{H}-^{13}\text{C}$ J coupling (~ 140 Hz)". We doubt the professionalism of this expert. First, the magnetic field of the nuclear magnetic resonance spectrometer (AVAVCE III HD400) should be ~ 400 MHz, rather than ~ 400 Hz. Second, the ~ 140 Hz of $^1\text{H}-^{13}\text{C}$ J coupling corresponding to ~ 0.35 ppm can occur definitely in the spectra. Additionally, the authors do not explain why all the ^1H NMR signals of the $^{13}\text{CH}_4$ and its oxidation products shift to high field in relation to the ^1H NMR signals of the natural CH_4 .

Author reply: We appreciate the reviewer' s insightful comments very much. We fully agreed with these comments and discussed with the faculty in charge of NMR facility in detail. After careful examinations of our previous ^1H NMR measurement, we found that the parameter of NMR was accidentally set as " ^1H experiment with decoupling", resulting the failure to give the $^1\text{H}-^{13}\text{C}$ coupling features.

During the course of revision, we re-measured the ^{13}C with decoupling NMR and ^1H NMR spectra of liquid-phase products of aqueous-phase photocatalytic conversion of methane over $\text{TiO}_2\{001\}-\text{C}_3\text{N}_4-0.1$ under the reaction condition of $4\%\text{O}_2+88\%\text{Ar}+165\ \mu\text{L}\ \text{H}_2\text{O}_2+20\ \text{mL}\ \text{H}_2\text{O}$ and of $8\%\text{CH}_4+4\%\text{O}_2+88\%\text{Ar}+165\ \mu\text{L}\ \text{H}_2\text{O}_2+20\ \text{mL}\ \text{H}_2\text{O}$ at 298 K using CH_4 or $^{13}\text{CH}_4$ at 298 K. The acquired ^1H NMR spectra show the ^1H - ^{13}C coupling features, consistent with those expected by the reviewer. These results, together with the MS results, confirm that all the products exclusively form from CH_4 .

We have revised Supplementary Figure 6 shown below in the revised Supplementary Materials and updated the file “Source data” accordingly.

Supplementary Figure 6. (a, b) ^1H NMR spectra of liquid-phase products of aqueous-phase photocatalytic conversion of methane over $\text{TiO}_2\{001\}-\text{C}_3\text{N}_4-0.1$ under the

reaction condition of 4%O₂+88%Ar+165 μL H₂O₂+20 mL H₂O and of 8%CH₄+4%O₂+88%Ar+165 μL H₂O₂+20 mL H₂O at 298 K using CH₄ or ¹³CH₄ at 298 K. The splitting peaks due to ¹H-¹³C J coupling for ¹³CH₃OH, ¹³CH₃OOH, H¹³COOH, ¹³CH₃¹³CH₂OH, and ¹³CH₃¹³COOH observed in these ¹H NMR spectra. (c) ¹³C with decoupling NMR spectra of liquid-phase products of aqueous-phase photocatalytic conversion of methane over TiO₂{001}-C₃N₄-0.1 under the reaction condition of 4%O₂+88%Ar+165 μL H₂O₂+20 mL H₂O and of 8%CH₄+4%O₂+88%Ar+165 μL H₂O₂+20 mL H₂O at 298 K using CH₄ or ¹³CH₄ at 298 K. (d) As-measured mass spectra of liquid-phase products of aqueous-phase photocatalytic conversion of methane over TiO₂{001}-C₃N₄-0.1 under the reaction condition of 8%CH₄+4%O₂+88%Ar+165 μL H₂O₂+20 mL H₂O at 298 K using ¹³CH₄ (top panel) or CH₄ (bottom panel) at 298 K. Photocatalyst amount: 20 mg; reaction time: 8 hours; stirring speed: 500 rpm.

We appreciate the reviewer's effort on reviewing our manuscript very much. We hope that the revised manuscript will be suitable for the publication in *Nature Communications*.

REVIEWERS' COMMENTS

Reviewer #3 (Remarks to the Author):

The authors answered all the issues from reviewers, it can be accepted by NC.

Author reply to Reviewer 3' s comments

Reviewer 3' s comments 3: The authors answered all the issues from reviewers, it can be accepted by NC.

Author reply: We appreciate the reviewer' s positive recommendation very much. We also appreciate the reviewer' s valuable comments very much which have greatly helped to improve the quality of our submission.